# LEARNING MIXED-CURVATURE REPRESENTATIONS IN PRODUCTS OF MODEL SPACES

**Albert Gu, Frederic Sala, Beliz Gunel & Christopher Ré**
Computer Science Department
Stanford University
Stanford, CA 94305
{`albertgu,fredsala,bgunel`}`@stanford.edu`, `chrismre@cs.stanford.edu`

## ABSTRACT

The quality of the representations achieved by embeddings is determined by how well the geometry of the embedding space matches the structure of the data. Euclidean space has been the workhorse for embeddings; recently hyperbolic and spherical spaces have gained popularity due to their ability to better embed new types of structured data—such as hierarchical data—but most data is not structured so uniformly. We address this problem by proposing learning embeddings in a product manifold combining multiple copies of these *model spaces* (spherical, hyperbolic, Euclidean), providing a space of heterogeneous curvature suitable for a wide variety of structures. We introduce a heuristic to estimate the sectional curvature of graph data and directly determine an appropriate signature—the number of component spaces and their dimensions—of the product manifold. Empirically, we jointly learn the curvature and the embedding in the product space via Riemannian optimization. We discuss how to define and compute intrinsic quantities such as means—a challenging notion for product manifolds—and provably learnable optimization functions. On a range of datasets and reconstruction tasks, our product space embeddings outperform single Euclidean or hyperbolic spaces used in previous works, reducing distortion by $32.55\%$ on a Facebook social network dataset. We learn word embeddings and find that a product of hyperbolic spaces in 50 dimensions consistently improves on baseline Euclidean and hyperbolic embeddings, by 2.6 points in Spearman rank correlation on similarity tasks and 3.4 points on analogy accuracy.

## 1 INTRODUCTION

With four decades of use, Euclidean space is the venerable elder of embedding spaces. Recently, non-Euclidean spaces—hyperbolic (Nickel & Kiela, 2017; Sala et al., 2018) and spherical (Wilson et al., 2014; Liu et al., 2017)—have gained attention by providing better representations for certain types of structured data. The resulting embeddings offer better reconstruction metrics: higher mean average precision (mAP) and lower distortion compared to their Euclidean counterparts. These three spaces are the *model spaces* of constant curvature (Lee, 1997), and this improvement in representation fidelity arises from the correspondence between the structure of the data (hierarchical, cyclical) and the geometry of non-Euclidean space (hyperbolic: negatively curved, spherical: positively curved). The notion of *curvature* plays the key role.

To improve representations for a variety of types of data—beyond hierarchical or cyclical—we seek spaces with heterogeneous curvature. The motivation for such *mixed* spaces is intuitive: our data may have complicated, varying structure, in some regions tree-like, in others cyclical, and we seek the best of all worlds. We expect mixed spaces to match the geometry of the data and thus provide higher quality representations. However, to employ these spaces, we face several key obstacles. We must perform a challenging manifold optimization to learn both the curvature and the embedding. Afterwards, we also wish to operate on the embedded points. For example, analogy operations for word embeddings in Euclidean vector space (e.g., $a - b + c$) must be lifted to manifolds.

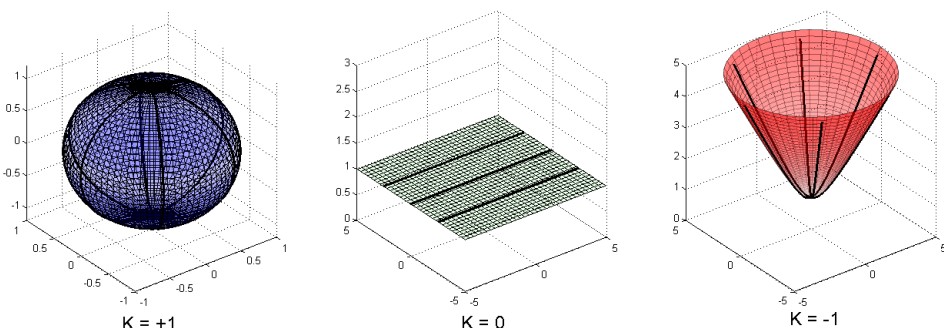

Figure 1: Three component spaces: sphere $\mathbb{S}^2$, Euclidean plane $\mathbb{E}^2$, and hyperboloid $\mathbb{H}^2$. Thick lines are geodesics; these get closer in positively curved ($K = +1$) space $\mathbb{S}^2$, remain equidistant in flat ($K = 0$) space $\mathbb{E}^2$, and get farther apart in negatively curved ($K = -1$) space $\mathbb{H}^2$.

We propose embedding into *product spaces* in which each component has constant curvature. As we show, this allows us to capture a wider range of curvatures than traditional embeddings, while retaining the ability to globally optimize and operate on the resulting embeddings. Specifically, we form a Riemannian product manifold combining hyperbolic, spherical, and Euclidean components and equip it with a decomposable Riemannian metric. While each component space in the product has constant curvature (positive for spherical, negative for hyperbolic, and zero for Euclidean), the resulting mixed space has *non-constant* curvature. However, selecting appropriate curvatures for the embedding space is a potential challenge. We directly learn the curvature for each component space along with the embedding (via Riemannian optimization), recovering the correct curvature, and thus the matching geometry, directly from data. We show empirically that we can indeed recover non-uniform curvatures and improve performance on reconstruction metrics.

Another technical challenge is to select the underlying number of components and dimensions of the product space; we call this the *signature*. This concept is vacuous in Euclidean space: the product of $\mathbb{E}^{r_1}, \ldots, \mathbb{E}^{r_n}$ is identical to the single space $\mathbb{E}^{r_1+\cdots+r_n}$. However, this is *not* the case with spherical and hyperbolic spaces. For example, the product of the spherical space $\mathbb{S}^1$ (the circle) with itself is the torus $\mathbb{S}^1 \times \mathbb{S}^1$, which is topologically distinct from the sphere $\mathbb{S}^2$. We address this challenge by introducing a theory-guided heuristic estimator for the signature. We do so by matching an empirical notion of discrete curvature in our data with the theoretical distribution of the sectional curvature, a fine-grained measure of curvature on Riemannian manifolds that is amenable to analysis in products. We verify that this approach recovers the correct signature on reconstruction tasks.

Standard techniques such as PCA require centering so that the embedded directions capture components of variation. Centering in turn needs an appropriate generalization of the mean. We develop a formulation of mean for embedded points that exploits the decomposability of the distance and has theoretical guarantees. For $T = \{p_1, \ldots, p_n\}$ in a manifold $\mathcal{M}$ with dimension $r$, the mean is $\mu(T) := \arg\min_p \sum_i d_{\mathcal{M}}^2(p, p_i)$. We give a global existence result: under symmetry conditions on the distribution of the points in $T$ on the spherical components, gradient descent recovers $\mu(T)$ with error $\varepsilon$ in time $O(nr \log \varepsilon^{-1})$.

We demonstrate the advantages of product space embeddings through a variety of experiments; products are at least as good as single spaces, but can offer significant improvements when applied to structures not suitable for single spaces. We measure reconstruction quality (via mAP and distortion) for synthetic and real datasets over various allocations of embedding spaces. We observe a $32.55\%$ improvement in distortion versus any single space on a Facebook social network graph. Beyond reconstruction, we apply product spaces to skip-gram word embeddings, a popular technique with numerous downstream applications, which crucially require the use of the manifold structure. We find that products of hyperbolic spaces improve performance on benchmark evaluations—suggesting that words form multiple smaller hierarchies rather than one larger one. We see an improvement of 3.4 points over baseline single spaces on the Google word analogy benchmark and of 2.6 points

in Spearman rank correlation on a word similarity task using the WS-353 corpus. Our results and initial exploration suggest that mixed product spaces are a promising area for future study.

## 2 PRELIMINARIES & BACKGROUND

**Embeddings**    For metric spaces[1] $U, V$ equipped with distances $d_U, d_V$, an *embedding* is a mapping $f : U \to V$. The quality of an embedding is measured by various *fidelity measures*. A standard measure is *average distortion* $D_{\text{avg}}$. The distortion of a pair of points $a, b$ is $(|d_V(f(a), f(b)) - d_U(a, b)|)/d_U(a, b)$, and $D_{\text{avg}}$ is the average over all pairs of points.

Distortion is a global metric; it considers the explicit value of all distances. At the other end of the global-local spectrum of fidelity measures is *mean average precision* (mAP), which applies to unweighted graphs. Let $G = (V, E)$ be a graph and node $a \in V$ have neighborhood $\mathcal{N}_a = \{b_1, \ldots, b_{\deg(a)}\}$, where $\deg(a)$ is the degree of $a$. In the embedding $f$, define $R_{a,b_i}$ to be the smallest ball around $f(a)$ that contains $b_i$ (that is, $R_{a,b_i}$ is the smallest set of nearest points required to retrieve the $i$th neighbor of $a$ in $f$). Then, $\text{mAP}(f) = \frac{1}{|V|} \sum_{a \in V} \frac{1}{\deg(a)} \sum_{i=1}^{|\mathcal{N}_a|} |\mathcal{N}_a \cap R_{a,b_i}|/|R_{a,b_i}|$.

Note that mAP does not track explicit distances; it is a ranking-based measure for local neighborhoods. Observe that $\text{mAP}(f) \leq 1$ (higher is better) while $d_{\text{avg}} \geq 0$ (lower is better).

**Riemannian Manifolds**    We briefly review some notions from manifolds and Riemannian geometry. A more in-depth treatment can be found in standard texts (Lee, 2012; do Carmo, 1992). Let $M$ be a smooth manifold, $p \in M$ be a point, and $T_p M$ be the tangent space to the point $p$. If $M$ is equipped with a Riemannian metric $g$, then the pair $(M, g)$ is called a *Riemannian manifold*. The shortest-distance paths on manifolds are called *geodesics*. To compute distance functions on a Riemannian manifold, the *metric tensor* $g$ is integrated along the geodesic. This is a smoothly varying function (in $p$) $g : T_p M \times T_p M \to \mathbb{R}$ that induces geometric notions such as length and angle by defining an inner product on the tangent space. For example, the norm of $v \in T_p M$ is defined as $\|v\|_g := g_p(v, v)^{\frac{1}{2}}$. In Euclidean space $\mathbb{R}^d$, each tangent space $T_p \mathbb{R}^d$ is canonically identified with $\mathbb{R}^d$, and the metric tensor $g^E$ is simply the normal inner product.

**Product Manifolds**    Consider a sequence of smooth manifolds $M_1, M_2, \ldots, M_k$. The product manifold is defined as the Cartesian product $M = M_1 \times M_2 \times \ldots \times M_k$. Notationally, we write points $p \in M$ through their coordinates $p = (p_1, \ldots, p_k) : p_i \in M_i$, and similarly a tangent vector $v \in T_p M$ can be written $(v_1, \ldots, v_k) : v_i \in T_{p_i} M_i$. If the $M_i$ are equipped with metric tensor $g_i$, then the product $M$ is also Riemannian with metric tensor $g(u, v) = \sum_{i=1}^k g_i(u_i, v_i)$. That is, the product metric decomposes into the sum of the constituent metrics.

**Geodesics and Distances**    Optimization on manifolds requires a notion of taking a step. This step can be performed in the tangent space and transferred to the manifold via the *exponential map* $\text{Exp}_p : T_p M \to M$. In a product manifold $\mathcal{P}$, for tangent vectors $v = (v_1, \ldots, v_k)$ at $p = (p_1, \ldots, p_k) \in M$, the exponential map simply decomposes, as do squared distances (Ficken, 1939; Turaga & Srivastava, 2016):

$$\text{Exp}_p(v) = (\text{Exp}_{p_1}(v_1), \ldots, \text{Exp}_{p_k}(v_k)), \qquad d_{\mathcal{P}}^2(x, y) = \sum_{i=1}^k d_i^2(x_i, y_i). \tag{1}$$

In other words, the shortest path between points in the product travels along the shortest paths in each component simultaneously. Note the analogy to Euclidean products $\mathbb{R}^d \equiv (\mathbb{R}^1)^d$.

**Hyperbolic and Spherical Model**    We use the *hyperboloid model* of hyperbolic space, with points in $\mathbb{R}^{d+1}$. Let $J \in \mathbb{R}^{(d+1) \times (d+1)}$ be the diagonal matrix with $J_{00} = -1$ and $J_{ii} = 1 : i > 0$. For $p, q \in \mathbb{R}^{d+1}$, the Minkowski inner product is $\langle p, q \rangle_* := p^T J q = -p_0 q_0 + p_1 q_1 + \ldots + p_d q_d$, and the corresponding norm is $\|p\|_* = \langle p, p \rangle_*^{\frac{1}{2}}$. For any $K > 0$, the hyperboloid $\mathbb{H}_K^d$ is defined on the

---

[1]In this paper, we use the language of graphs; note that any discrete metric space can be identified with a weighted graph, and all of our algorithms operate on weighted graphs.

subset $\{p \in \mathbb{R}^{d+1} : \|p\|_* = -K^{1/2}, p_0 > 0\}$. When the subscript $K$ is omitted, it is taken to be 1. The hyperbolic distance on $\mathbb{H}^d$ is $d_H(p, q) = \text{acosh}(-\langle p, q \rangle_*)$.

Similarly, spherical space $\mathbb{S}_K^d$ is most easily defined when embedded in $\mathbb{R}^{d+1}$. The manifold is defined on the subset $\{p \in \mathbb{R}^{d+1} : \|p\|_2 = K^{1/2}\}$, with metric $g^S$ induced by the Euclidean metric on $\mathbb{R}^{d+1}$. The spherical distance on $\mathbb{S}^d$ is $d_S(p, q) = \arccos(\langle p, q \rangle)$.

## 3 Product Spaces and Constructions

We now tackle the challenges of mixed spaces. First, we introduce a product manifold embedding space $\mathcal{P}$ composed of multiple copies of simple model spaces, providing heterogeneous curvature. Next, in Section 3.1, given the signature of $\mathcal{P}$ (the number of components of each type and their dimensions), we describe how to simultaneously learn an embedding and the curvature for each component through optimization. In Section 3.2, we provide a heuristic to choose the signature by estimating a discrete notion of curvature for given data. Finally, in Section 3.3, given an embedding in $\mathcal{P}$, we introduce a Karcher-style mean which can be recovered efficiently.

Let $\mathbb{S}_K^d$ and $\mathbb{H}_K^d$ be the spherical and hyperbolic spaces of dimension $d$ and curvature $K, -K$, respectively, and $\mathbb{E}^d$ the Euclidean space of dimension $d$.[2] We describe our main embedding space: for sequences of dimensions $s_1, s_2, \ldots, s_m, h_1, \ldots, h_n$, and $e$, we write

$$\mathcal{P} = \mathbb{S}^{s_1} \times \mathbb{S}^{s_2} \times \cdots \times \mathbb{S}^{s_m} \times \mathbb{H}^{h_1} \times \mathbb{H}^{h_2} \times \cdots \times \mathbb{H}^{h_n} \times \mathbb{E}^e,$$

a product manifold with $m + n + 1$ component spaces and total dimension $\sum_i s_i + \sum_j h_j + e$. We refer to each $\mathbb{S}^{s_i}, \mathbb{H}^{h_i}, \mathbb{E}^e$ as *components* or *factors*. We refer to the decomposition, e.g., $(\mathbb{H}^2)^2 = \mathbb{H}^2 \times \mathbb{H}^2$, as the *signature*. For convenience, let $M_1, \ldots, M_{m+n+1}$ refer to the factors in the product.

**Distances on $\mathcal{P}$** As discussed in Section 2, the product $\mathcal{P}$ is a Riemannian manifold defined by the structure of its components. For $p, q \in \mathcal{P}$, we write $d_{M_i}(p, q)$ for the distance $d_{M_i}$ restricted to the appropriate components of $p$ and $q$ in the product. In particular, the squared distance in the product decomposes via (1). In other words, $d_{\mathcal{P}}$ is simply the $\ell_2$ norm of the component distances $d_{M_i}$.

We note that $\mathcal{P}$ can also be equipped with different distances (ignoring the Riemannian structure), leading to a different embedding space. Without the underlying manifold structure, we cannot freely operate on the embedded points such as taking geodesics and means, but some simple applications only interact through distances. For such settings, we consider the $\ell_1$ distance $d_{\mathcal{P},\ell_1}(p, q) = \sum_{i=1}^{s_m} d_{S_i}(p, q) + \sum_{i=1}^{h_n} d_{H_i}(p, q) + d_E(p, q)$ and the min distance $d_{\mathcal{P},\min}(p, q) = \min\{d_{S_1}(p, q), \ldots, d_{H_1}(p, q), \ldots, d_E(p, q)\}$. These distances provide simple and interpretable embedding spaces using $\mathcal{P}$, enabling us to introduce *combinatorial constructions* that allow for embeddings without the need for optimization. We give an example below and discuss further in the Appendix. We then focus on the Riemannian distance, which allows Riemannian optimization directly on the manifold, and enables full use of the manifold structure in generic downstream applications.

**Example** Consider the graph $G$ shown on the right of Figure 2. This graph has a backbone cycle with 9 nodes, each attached to a tree; such topologies are common in networking. If a single edge $(a, b)$ is removed from the cycle, the result is a tree embeddable arbitrarily well into hyperbolic space (Sala et al., 2018). However, $a, b$ (and their subtrees) would then incur an additional distance of $8 - 1 = 7$, being forced to go the other way around the cycle. But using the $\ell_1$ distance, we can embed $G_{\text{tree}}$ into $\mathbb{H}^2$ and $G_{\text{cycle}}$ into $\mathbb{S}^1$, yielding arbitrarily low distortion for $G$. We give the full details and another combinatorial construction for the min-distance in the Appendix.

### 3.1 Optimization & Component Curvatures

To compute embeddings, we optimize the placement of points through an auxiliary loss function. Given graph distances $\{d_G(X_i, X_j)\}_{ij}$, our loss function of choice is

$$\mathcal{L}(x) = \sum_{1 \leq i \leq j \leq n} \left| \left( \frac{d_{\mathcal{P}}(x_i, x_j)}{d_G(X_i, X_j)} \right)^2 - 1 \right|, \tag{2}$$

---

[2]We write $\mathbb{E}$ for our Euclidean embedding space component to distinguish it from $\mathbb{R}$, since our models of hyperbolic and spherical geometry also use $\mathbb{R}$ as an ambient space.

**Algorithm 1** R-SGD in products

1: **Input: Loss function** $L : \mathcal{P} \to \mathbb{R}$
2: Initialize $x^{(0)} \in \mathcal{P}$ randomly
3: **for** $t = 0, \ldots, T - 1$ **do**
4:     $h \leftarrow \nabla L(x^{(t)})$
5:     **for** $i = 1, \ldots, m$ **do**
6:         $v_i \leftarrow \mathrm{proj}_{x_i^{(t)}}^S(h_i)$
7:     **for** $i = m + 1, \ldots, m + n$ **do**
8:         $v_i \leftarrow \mathrm{proj}_{x_i^{(t)}}^H(h_i)$
9:         $v_i \leftarrow J v_i$
10:     $v_{m+n+1} \leftarrow h_{m+n+1}$
11:     **for** $i = 1, \ldots, m + n + 1$ **do**
12:         $x_i^{(t+1)} \leftarrow \mathrm{Exp}_{x_i^{(t)}}(v_i)$
13: **return** $x^{(T)}$

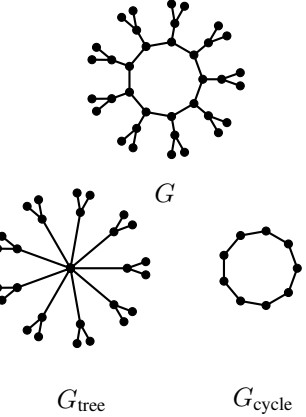

Figure 2: Left: Riemannian SGD decomposes per component. Subscripts $i$ index components in the product. Right: Ring of trees graph $G$. Neither hyperbolic nor spherical space is suitable for $G$, but the product $\mathbb{H} \times \mathbb{S}$ captures it with low distortion. Note the decomposition into tree and cycle.

which captures the average distortion. (2) depends on hyperbolic distance $d_H$ (for which the gradient is unstable) only through the square $d_H^2$, which is continuously differentiable (Sala et al., 2018).

In any Riemannian manifold, a loss function can be optimized through standard Riemannian optimization methods such as RSGD (Bonnabel, 2013) and RSVRG (Zhang et al., 2016). We write down the full RSGD specialized to our product spaces in Algorithm 1. This proceeds by first computing the Euclidean gradient $\nabla L(x)$ with respect to the ambient space of the embedding (Step 4), and then converting it to the Riemannian gradient by applying the Riemannian correction (multiply by the inverse of the metric tensor $g_{\mathcal{P}}^{-1}$). This overall strategy has been detailed in previous work in the hyperboloid model (Nickel & Kiela, 2018; Wilson & Leimeister, 2018), and the same calculations apply to our hyperbolic components.

Since $g_{\mathcal{P}}$ is block diagonal on a product manifold, it suffices to apply the correction and perform the gradient step in each component $M_i$ independently. In the spherical and hyperboloid models, which have smaller dimension than the ambient space, this is performed by first projecting the gradient vector $h$ onto the tangent space $T_x M$ via $\mathrm{proj}_x^S(h) = h - \langle h, x \rangle x$ (Step 6) and $\mathrm{proj}_x^H(h) = h + \langle h, x \rangle_* x$ (Step 8). In the hyperboloid model, a final rescaling by the inverse of the metric $J$ is needed (Step 9). This is not required in the spherical model since it inherits the same metric from the ambient Euclidean space.

**Learning the Curvature**   There exists a spherical model for every curvature $K > 0$ (for example, the sphere $\mathbb{S}_K^d$ of radius $K^{-1/2}$) and a hyperbolic model for every $K < 0$ (the hyperboloid $\mathbb{H}_{-K}^d$). We jointly optimize the curvature $K_i$ of every non-Euclidean factor $M_i$ along with the embeddings.

The idea is that distances on the spherical and hyperboloid models of arbitrary curvature can be emulated through distances on the standard models $\mathbb{S}, \mathbb{H}$ of curvature 1. For example, given $p, q$ on the sphere $\mathbb{S}_{1/R^2}$ of radius $R$, then $d(p, q) = R \cdot d_{\mathbb{S}_1}(p/R, q/R)$ where $p/R, q/R$ lie on the unit sphere. Therefore the radius $R$, which is monotone in the curvature $K$, can be treated as a parameter as well, so that we can optimize $K$ and implicitly represent points lying on the manifold of curvature $K$, while explicitly only needing to store and optimize points in the standard model of curvature 1 via Algorithm 1. The hyperboloid model is analogous. Moreover, the loss (2) depends only on squared distances on the product manifold, which are simple functions of distances in the components through (1), so we can optimize the curvature of each factor in $\mathcal{P}$.

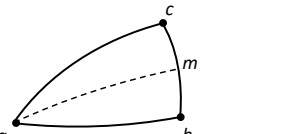 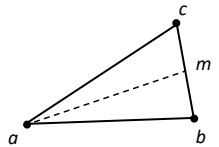 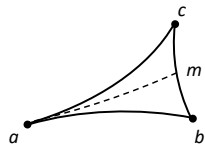

Figure 3: Geodesic triangles in differently curved spaces: compared to Euclidean geometry in which it satisfies the parallelogram law (Center), the median $am$ is longer in cycle-like positively curved space (Left), and shorter in tree-like negatively curved space (Right). The relative length of $am$ can be used as a heuristic to estimate discrete curvature.

## 3.2 ESTIMATING THE SIGNATURE

To choose the signature of an appropriate space $\mathcal{P}$ corresponding to given data, we again turn to curvature. We use the *sectional curvature*, a finer-grained notion defined over all two-dimensional subspaces passing through a point. Unlike coarser notions like scalar curvature, this is not constant in a product of basic spaces. Given linearly independent $u, v \in T_p M$ spanning a two-dimensional subspace $U$, the sectional curvature $K_p(u, v)$ or $K_p(U)$ is defined as the Gaussian curvature of the surface $\mathrm{Exp}(U) \subseteq M$. Intuitively, this captures the rate that geodesics on the surface emanating from $p$ spread apart, which relates to volume growth. In Appendix C.2, we show that the sectional curvature of $\mathcal{P}$ interpolates between the sectional curvatures of the factors, enabling us to better capture a wider range of structures in our embeddings:

**Lemma 1.** *Let $M = M_1 \times M_2$ where $M_i$ has constant curvature $K_i$. For any $u, v \in T_p M$, if $K_1, K_2$ are both non-negative, the sectional curvature satisfies $K(u, v) \in [0, \max\{K_1, K_2\}]$. If $K_1, K_2$ are both non-positive, the sectional curvature satisfies $K(u, v) \in [\min\{K_1, K_2\}, 0]$. If $K_i < 0$ and $K_j > 0$ for $i \neq j$, then $K(u, v) \in [K_i, K_j]$.*

Our estimation technique employs a triangle comparison theorem following from Toponogov's theorem and the law of cosines, which characterizes sectional curvature through the behavior of small triangles (note that a triangle determines a 2-dimensional submanifold). Let $abc$ be a geodesic triangle in manifold (or metric space) $M$ and $m$ be the (geodesic) midpoint of $bc$, and consider the quantity

$$\xi_M(a, b, c) := d_M(a, m)^2 + d_M(b, c)^2/4 - \left(d_M(a, b)^2 + d_M(a, c)^2\right)/2. \tag{3}$$

This is non-negative (resp. non-positive) when the curvature is non-negative (resp. non-positive). Note that consequently the equality case occurs exactly when the curvature is 0, and equation 3 becomes the *parallelogram law* of Euclidean geometry (Figure 3).

Analogous to sectional curvature, which is a function of a point $p$ and two directions $x, y$ from $p$, in an undirected graph $G$ we define an analog for every node $m$ and two neighbors $b, c$. Given a reference node $a$ we set: $\xi_G(m; b, c; a) = \frac{1}{2d_G(a,m)}\xi_G(a, b, c)$. This is exactly the expression from equation 3, normalized suitably so as to yield the correct scaling for trees and cycles. Our curvature estimation is then a simple average $\xi_G(m; b, c) = \frac{1}{|V|-1}\sum_{a \neq m}\xi_G(m; b, c; a)$.

Importantly, $\xi_G$ recovers the right curvature for graph atoms such as lines, cycles, and trees (Appendix C.2, Lemma 4,5), and the correct sign for other special discrete objects like polyhedra (Thurston, 1998). The curvature is zero for lines, positive for cycles, and negative for trees.

For a generic graph $G$, we use this to generate a potential product manifold to embed in. An empirical sectional curvature of $G$ is estimated via Algorithm 3, which is based off the *homogeneity* of product manifolds (i.e. isometries act transitively), implying that it suffices to analyze the curvature at a random point. In particular, we moment-match the distributions of sectional curvature through uniformly random 2-planes in the graph and in the manifold through Algorithms 3,2 (Appendix C.2).

## 3.3 MEANS IN THE PRODUCT MANIFOLD

A critical operation on manifolds is that of *taking the mean*; it is necessary for many downstream applications, including, for example, analogy tasks with word embeddings, for clustering, and for

centering before applying PCA. Even in simple settings like the circle $\mathbb{S}^1$, defining a mean is non-trivial. A classic approach is to take the Euclidean mean (in $\mathbb{E}^2$) of the points and to project back onto $\mathbb{S}^1$—but this operation fails in the case where the points are uniformly spaced on $\mathbb{S}^1$. A further roadblock is the varying curvature of $\mathcal{P}$. Fortunately, we can exploit the decomposability of the distance on $\mathcal{P}$, reducing the challenge to breaking symmetries in the component spaces. To do so, we introduce the following Karcher-style weighted mean. Let $T = \{p_1, p_2, \ldots, p_n\}$ be a set of points in $\mathcal{P}$ and $w_1, \ldots, w_n$ be positive weights satisfying $\sum_{i=1}^{n} w_i = 1$. Then the mean $\mu(T)$ is $\arg\min_{p \in \mathcal{P}} \sum_{i=1}^{n} w_i d_{\mathcal{P}}^2(p, p_i)$. In special cases, this matches commonly used means (the centroid in the Euclidean case $\mathbb{E}^d$, the spherical average for $\mathbb{S}^2$ in Buss & Fillmore (2001)). We further note that when $w_i \geq 0$, the squared-distance components above are individually convex: this is trivial in the Euclidean term, holds in the hyperbolic case (cf. Theorem 4.1 (Bishop & O'Neill, 1969)), and holds in the spherical case under certain restrictions, e.g., when the points in $T$ lie entirely in one hemisphere of $\mathbb{S}^r$ (Buss & Fillmore, 2001). Moreover, in this case, peforming the optimization on the mean with gradient descent via the exponential map offers linear rate convergence:

**Lemma 2.** *Let $\mathcal{P}$ be a product of model spaces of total dimension $r$, $T = \{p_1, \ldots, p_n\}$ points in $\mathcal{P}$ and $w_1, \ldots, w_n$ weights satisfying $w_i \geq 0$ and $\sum_{i=1}^{n} w_i = 1$. Moreover, let the components of the points in $\mathcal{P}$, $p_{i|\mathbb{S}^j}$ restricted to each spherical component space $\mathbb{S}^j$ fall in one hemisphere of $\mathbb{S}^j$. Then, Riemannian gradient descent recovers the mean $\mu(T)$ within distance $\epsilon$ in time $O(nr \log \epsilon^{-1})$.*

This is a *global* result; with weaker assumptions, we can derive local results; for example, in the case where some of the $w_i$ are negative, which is useful for analogy operations.

In summary, we offer the following key takeaways of our development:

- Product manifolds of model spaces capture heterogeneous curvature while providing tractable optimization,

- Each component's curvature can be learned empirically through a reparametrization,

- A signature for the product can be found by matching discrete notions of curvature on graphs with sectional curvature on manifolds,

- There exists an easily-computed formulation of mean with theoretical guarantees.

## 4 EXPERIMENTS

We evaluate the proposed approach, comparing the representation quality of synthetic graphs and real datasets among different embedding spaces by measuring the reconstruction fidelity (through average distortion and mAP). We expect that mixed product spaces perform better for non-homogeneous data. We consider the curvature of graphs, reporting the curvatures learned through optimization as well as the theoretical allocation from Section 3.2. Beyond reconstruction, we evaluate the intrinsic performance of product space embeddings in a skip-gram word embedding model, by defining tasks with generic manifold operations such as means.

### 4.1 GRAPH RECONSTRUCTION

**Datasets** We examine synthetic datasets—trees, cycles, the ring of trees shown in Figure 1, confirming that each matches its theoretically optimal embedding space. We then compare on several real-world datasets with describable structure, including the USCA312 dataset of distances between North American cities (Burkardt); a tree-like graph of computer science Ph.D. advisor-advisee relationships (De Nooy et al., 2011) reported in previous hyperbolics work (Sala et al., 2018); a power-grid distribution network with backbone structure (Watts & Strogatz, 1998); and a dense social network from Facebook (McAuley & Leskovec, 2012). For the former two graphs with well-defined structure, we expect optimal embeddings in spaces of positive and negative curvature, respectively. We hypothesize that the backbone network embeds well into simple products of hyperbolic and spherical spaces as in Figure 2, and the dense graph also benefits from a mixture of spaces.

**Approaches** We minimize the loss (2) using Algorithm 1. We fix a total dimension $d$ and consider the most natural ways to construct product manifolds of the given dimension, through iteratively

Table 1: **Matching geometries**: Average distortion on canonical graphs (tree, cycle, ring of trees) with 40 nodes, comparing four spaces with total dimension 3. The best distortion is achieved by the space with matching geometry.

|  | **Cycle** | **Tree** | **Ring of Trees** |
|---|---|---|---|
|  | $\|V\| = 40, \|E\| = 40$ | $\|V\| = 40, \|E\| = 39$ | $\|V\| = 40, \|E\| = 40$ |
| $(\mathbb{E}^3)^1$ | 0.1064 | 0.1483 | 0.0997 |
| $(\mathbb{H}^3)^1$ | 0.1638 | **0.0321** | 0.0774 |
| $(\mathbb{S}^3)^1$ | **0.0007** | 0.1605 | 0.1106 |
| $(\mathbb{H}^2)^1 \times (\mathbb{S}^1)^1$ | 0.1108 | 0.0538 | **0.0616** |

doubling the number of factors. These models include the products consisting of only a constant-curvature base space, ranging to various combinations of $\mathbb{S}_2^{d/2}, \mathbb{H}_2^{d/2}$ comprising factors of dimension 2.[3] For a given signature, the curvatures are initialized to the appropriate value in $\{-1, 0, 1\}$ and then learned using the technique in Section 3.1. We additionally compare to the outputs of Algorithms 2,3 for heuristically selecting a combination of spaces in which to embed these datasets.

**Quality**   We focus on the average distortion—which our loss function (2) optimizes—as our main metric for reconstruction, and additionally report the mAP metric for the unweighted graphs. As expected, for the synthetic graphs (tree, cycle, ring of trees), the matching geometries (hyperbolic, spherical, product of hyperbolic and spherical) yield the best distortion (Table 1). Next, we report in Table 2 the quality of embedding different graphs across a variety of allocations of spaces, fixing total dimension $d = 10$ following previous work (Nickel & Kiela, 2018). We confirm that the structure of each graph informs the best allocation of spaces. In particular, the cities graph—which has intrinsic structure close to $\mathbb{S}^2$—embeds well into any space with a spherical component, and the tree-like Ph.D.s graph embeds well into hyperbolic products. We emphasize that even for such datasets that theoretically match a single constant-curvature space, the products thereof perform no worse. In general, the product construction achieves high quality reconstruction: the traditional Euclidean approach is often well below several other signatures. We additionally report the learned curvatures associated with the optimal signature, finding that the resulting curvatures are non-uniform even for products of identical spaces (cf. Ph.D.s). Finally, Table 3 reports the signature estimations of Algorithms 2, 3 for the unweighted graphs. Among the signatures over two components, the estimated curvature signs agree with best distortion results from Table 2.

### 4.2   WORD EMBEDDINGS

To investigate the performance of product space embeddings in applications requiring the underlying manifold structure, we learned word embeddings and evaluated them on benchmark datasets for word similarity and analogy. In particular, we extend results on hyperbolic skip-gram embeddings from Leimeister & Wilson (2018) (LW), who found that hyperbolic embeddings perform favorably against Euclidean word vectors in low dimensions ($d = 5, 20$), but less so in higher dimensions ($d = 50, 100$). Building on these results, we hypothesize that in high dimensions, a product of multiple smaller-dimension hyperbolic spaces will substantially improve performance.

**Setup**   We use the standard skip-gram model (Mikolov et al., 2013) and extend the loss function to a generic objective suitable for arbitrary manifolds, which is a variant of the objective proposed by LW. Concretely, given a word $u$ and target $w$, with label $y = 1$ if $w$ is a context word for $u$ and $y = 0$ if it is a negative sample, the model is $P(y|w, u) = \sigma \left( (-1)^{1-y} (-\cosh(d(\alpha_u, \gamma_w)) + \theta) \right)$.

Training followed the setup of LW, building on the *fastText* skip-gram implementation. Euclidean results are reported directly from *fastText*. Aside from choice of model, the training setup including hyperparameters (window size, negative samples, etc.) is identical to LW for all models.

**Word Similarity**   We measure the Spearman rank correlation $\rho$ between our scores and annotated ratings on the word similarity datasets WS-353 (Finkelstein et al., 2001), Simlex-999 (Hill et al.,

---

[3]Note that $\mathbb{S}^1$ and $\mathbb{H}^1$ are metrically equivalent to $\mathbb{R}$, so these are not considered.

Table 2: **Graph reconstruction**: fidelity measures for graph embeddings using $d = 10$ total dimensions, with varying allocations of spaces and dimensions. Our loss function (2) targets distortion, and for each dataset the best model reflects the structure of the data. Even on near-perfectly spherical or hierarchical data, products of $\mathbb{S}$ (resp. $\mathbb{H}$) perform no worse than the single copy.

| | **Cities** | **CS PhDs** | | **Power** | | **Facebook** | |
|---|---|---|---|---|---|---|---|
| | $|V|=312$ | $|V|=1025, |E|=1043$ | | $|V|=4941, |E|=6594$ | | $|V|=4039, |E|=88234$ | |
| | $D_{\text{avg}}$ | $D_{\text{avg}}$ | mAP | $D_{\text{avg}}$ | mAP | $D_{\text{avg}}$ | mAP |
| $\mathbb{E}^{10}$ | 0.0735 | 0.0543 | 0.8691 | 0.0917 | 0.8860 | 0.0653 | 0.5801 |
| $\mathbb{H}^{10}$ | 0.0932 | 0.0502 | 0.9310 | 0.0388 | 0.8442 | 0.0596 | 0.7824 |
| $\mathbb{S}^{10}$ | 0.0598 | 0.0569 | 0.8329 | 0.0500 | 0.7952 | 0.0661 | 0.5562 |
| $(\mathbb{H}^5)^2$ | 0.0756 | 0.0382 | 0.9628 | 0.0365 | 0.8605 | 0.0430 | 0.7742 |
| $(\mathbb{S}^5)^2$ | **0.0593** | 0.0579 | 0.7940 | 0.0471 | 0.8059 | 0.0658 | 0.5728 |
| $\mathbb{H}^5 \times \mathbb{S}^5$ | 0.0622 | 0.0509 | 0.9141 | **0.0323** | 0.8850 | **0.0402** | 0.7414 |
| $(\mathbb{H}^2)^5$ | 0.0687 | **0.0357** | 0.9694 | 0.0396 | 0.8739 | 0.0525 | 0.7519 |
| $(\mathbb{S}^2)^5$ | 0.0638 | 0.0570 | 0.8334 | 0.0483 | 0.8818 | 0.0631 | 0.5808 |
| $(\mathbb{H}^2)^2 \times \mathbb{E}^2 \times (\mathbb{S}^2)^2$ | 0.0765 | 0.0391 | 0.8672 | 0.0380 | 0.8152 | 0.0474 | 0.5951 |
| **Best model** | $\mathbb{S}^5_{1.0} \times \mathbb{S}^5_{1.1}$ | $\mathbb{H}^2_{.3} \times \mathbb{H}^2_{.6} \times \mathbb{H}^2_{1.5} \times (\mathbb{H}^2_{1.2})^2$ | | $\mathbb{H}^5_{3.4} \times \mathbb{S}^5_{12.6}$ | | $\mathbb{H}^5_{0.3} \times \mathbb{S}^5_{3.5}$ | |
| $D_{\text{avg}}$ **improvement over single space** | 0.8% | 28.89% | | 16.75% | | 32.55% | |

Table 3: **Heuristic allocation:** estimated signatures for embedding unweighted graphs from Table 2 into two factors, using Algorithms 2,3 to match the empirical distribution of graph curvature. The resulting curvature signs agree with results from Table 2 for choosing among two-component spaces.

| | CS PhDs | Power | Facebook |
|---|---|---|---|
| Estimated Signature | $\mathbb{H}^5_{1.3} \times \mathbb{H}^5_{0.2}$ | $\mathbb{H}^5_{1.8} \times \mathbb{S}^5_{1.7}$ | $\mathbb{H}^5_{0.9} \times \mathbb{S}^5_{1.6}$ |

2015) and MEN (Bruni et al., 2014). The results are in Table 4. Notably, we find that hyperbolic word embeddings are consistently competitive with or better than the Euclidean embeddings, and the improvement increases with more factors in the product. This suggests that word embeddings implicitly contain multiple distinct but smaller hierarchies rather than forming a single larger one.

**Analogies**  In manifolds, there is no exact analog of the "word arithmetic" of conventional word embeddings arising from vector space structure. However, analogies can still be defined via intrinsic product manifold operations. In particular, note that the loss function depends on the embeddings solely through their pairwise distances. We thus define analogies $a : b :: c : d$ by matching the distances $d^2(a, b) = d^2(c, d)$ and $d^2(a, c) = d^2(b, d)$ through constructing an analog of the parallelogram, by geodesically reflecting $a$ through the geodesic midpoint (i.e. mean) $m$ of $b, c$. Note that this defines both the loss function and the intrinsic tasks purely in terms of distances and manifold operations. Hence, unlike traditional word embeddings, this formulation is generic to any space.

Our evaluation, shown in Table 5, uses the standard Google word analogy benchmark (Mikolov et al., 2013). We observe a 22% accuracy improvement over single-space hyperbolic embeddings in 50 dimensions and similar improvements over a single hyperbolic space in 100 dimensions. As with similarity, accuracy on the analogy task consistently improves as the number of factors increases.

## 5 CONCLUSION

Product spaces enable improved representations by better matching the geometry of the embedding space to the structure of the data. We introduced a tractable Riemannian product manifold class that combines Euclidean, spherical, and hyperbolic spaces. We showed how to learn embeddings and curvatures, estimate the product signature, and defined a tractable formulation of mean. We hope that our techniques encourage further research on non-Euclidean embedding spaces.

Table 4: Spearman rank correlation on similarity datasets. Top: Previous results from embeddings into spaces of fixed curvature. Bottom: Embeddings into products of $\mathbb{H}$ with fixed total dimension.

|  | Dim 50 | | | Dim 100 | | |
|---|---|---|---|---|---|---|
|  | WS-353 | Simlex | MEN | WS-353 | Simlex | MEN |
| Euclidean | 0.6628 | 0.2738 | 0.7217 | 0.6986 | 0.2923 | 0.7473 |
| Hyperbolic | 0.6787 | 0.2784 | 0.7117 | 0.6846 | 0.2832 | 0.7217 |
| 2 Hyperbolics | 0.6955 | **0.2870** | 0.7246 | 0.7297 | 0.3168 | 0.7450 |
| 5 Hyperbolics | **0.7048** | 0.2837 | **0.7270** | **0.7379** | **0.3212** | **0.7530** |

Table 5: Accuracy on the Google word analogy dataset. Taking products of smaller hyperbolic spaces significantly improves performance. Unlike conventional embeddings, the operations in hyperbolic and product spaces are defined solely through distances and manifold operations.

| Total Dim $d$ / Model | $\mathbb{R}^d$ | $(\mathbb{H}^d)^1$ | $(\mathbb{H}^{d/2})^2$ | $(\mathbb{H}^{d/5})^5$ | $(\mathbb{H}^2)^{d/2}$ |
|---|---|---|---|---|---|
| 50 | 0.3866 | 0.3424 | 0.3928 | 0.4181 | **0.4209** |
| 100 | **0.5513** | 0.3738 | 0.4310 | 0.4731 | 0.5216 |

## ACKNOWLEDGMENTS

We gratefully acknowledge the support of DARPA under Nos. FA87501720095 (D3M) and FA86501827865 (SDH), NIH under No. N000141712266 (Mobilize), NSF under Nos. CCF1763315 (Beyond Sparsity) and CCF1563078 (Volume to Velocity), ONR under No. N000141712266 (Unifying Weak Supervision), the Moore Foundation, NXP, Xilinx, LETI-CEA, Intel, Google, NEC, Toshiba, TSMC, ARM, Hitachi, BASF, Accenture, Ericsson, Qualcomm, Analog Devices, the Okawa Foundation, and American Family Insurance, and members of the Stanford DAWN project: Intel, Microsoft, Teradata, Facebook, Google, Ant Financial, NEC, SAP, and VMWare. The U.S. Government is authorized to reproduce and distribute reprints for Governmental purposes notwithstanding any copyright notation thereon. Any opinions, findings, and conclusions or recommendations expressed in this material are those of the authors and do not necessarily reflect the views, policies, or endorsements, either expressed or implied, of DARPA, NIH, ONR, or the U.S. Government.

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

The Appendix starts with a glossary of symbols and a discussion of related work. Afterwards, we provide the proof of Lemma 2. We continue with a more in-depth treatment of the curvature estimation algorithm. We then introduce two combinatorial constructions—embedding techniques that do not require optimization—that rely on the alternative product distances. We give additional details on our experimental setup. Finally, we additionally evaluate the interpretability of these embeddings (i.e., do the separate components in the embedding manifold capture intrinsic qualities of the data?) through visualizations of the synthetic example from Figure 1.

## A  GLOSSARY OF SYMBOLS

We provide a glossary of commonly-used terms in our paper.

| Symbol | Used for |
|---|---|
| $\mathrm{mAP}(f)$ | the mean average precision fidelity measure of the embedding $f$ |
| $D(f)$ | the distortion fidelity measure of the embedding $f$ |
| $D_{\mathrm{wc}}(f)$ | the worst-case distortion fidelity measure of the embedding $f$ |
| $G$ | a graph, typically with node set $V$ and edge set $E$ |
| $T$ | a tree |
| $a, b, c$ | nodes in a graph or tree |
| $f$ | an embedding |
| $\mathcal{N}_a$ | neighborhood around node $a$ in a graph |
| $R_{a,b}$ | the smallest set of closest points to node $a$ in an embedding $f$ that contains node $b$ |
| $M$ | a manifold; when equipped with a metric $g$, $M$ is Riemannian |
| $p$ | a point in a manifold, $p \in M$ |
| $T_p M$ | the tangent space of point $p$ in $M$ (a vector space) |
| $g$ | a Riemannian metric defining an inner product on $T_p M$ |
| $\mathbb{E}^d$ | $d$-dimensional Euclidean space |
| $\mathbb{S}^d$ | $d$-dimensional spherical space |
| $\mathbb{H}^d$ | $d$-dimensional hyperbolic space |
| $\mathcal{P}$ | product manifold consisting of spherical, Euclidean, hyperbolic factors |
| $\mathrm{Exp}_x(v)$ | the exponential map for tangent vector $v$ at point $x$ |
| $R$ | the Riemannian curvature tensor |
| $K(x, y)$ | the sectional curvature for a subspace spanned by linearly independent $x, y \in T_p M$ |
| $d_E$ | metric distance between two points in Euclidean space |
| $d_S$ | metric distance between two points in spherical space |
| $d_H$ | metric distance between two points in hyperbolic space |
| $d_U$ | metric distance between two points in metric space $U$ |
| $d_G$ | metric distance between two points in a graph $G = (V, E)$ |
| $\mu(T)$ | mean of a set of points $T = \{p_1, \ldots, p_n\}$ in $\mathcal{P}$ |
| $\mathbb{I}_n$ | the $n \times n$ identity matrix |

Table 6: Glossary of variables and symbols used in this paper.

## B  RELATED WORK

Hyperbolic space has recently been proposed as an alternative to Euclidean space to learn embeddings in cases where there is a (possibly latent) hierarchical structure. In fact, many types of data (from various domains) such as social networks, word frequencies, metabolic-mass relationships, and phylogenetic trees of DNA sequences exhibit a non-Euclidean latent structure, as shown in Bronstein et al. (2017).

Initial works on hyperbolic embeddings include Nickel & Kiela (2017) and Chamberlain et al. (2017). In Chamberlain et al. (2017), neural graph embeddings are performed in hyperbolic space and used to classify the vertices of complex networks. A similar application is link prediction in Nickel & Kiela (2017) for the lexical database WordNet; this work also measured predicted lexical entailment on the HyperLex benchmark dataset. The follow-up work Nickel & Kiela (2018) performs optimizations in the hyperboloid (i.e. Lorentz) model instead of the Poincaré model.

Tay et al. (2018) proposed a neural ranking based question answering (Q/A) system in hyperbolic space that outperformed many state-of-the-art models using fewer parameters compared to competitor learning models. Ganea et al. (2018a) proposed hyperbolic embeddings of entailment relations, described by directed acyclic graphs by applying hyperbolic cones as a heuristic and showed improvements over baselines in terms of representational capacity and generalization. Sala et al. (2018) developed a combinatorial construction for efficiently embedding trees and tree-like graphs without optimization, studied the fundamental tradeoffs of hyperbolic embeddings, and explored PCA-like algorithms in hyperbolic space.

Unlike Euclidean space, most Riemannian manifolds are not vector spaces, and thus even basic operations such as vector addition, vector translation and matrix multiplication do not have universal interpretations. In more complex geometries, closed form expressions for basic objects like distances, geodesics, and parallel transport do not exist. As a result, standard machine learning or deep learning tools, such as convolutional neural networks, long short term memory networks (LSTMs), logistic regression, support vector machines, and attention mechanisms, do not have exact correspondences in these complex geometries.

A pair of recent approaches seek to formulate standard machine learning methods in hyperbolic space. Gulcehre et al. (2018) introduces a hyperbolic version of the attention mechanism using the hyperboloid model. This work shows improvements in terms of generalization on several downstream applications including neural machine translation, learning on graphs and visual question answering tasks, while having compact neural representations. Ganea et al. (2018b) formulates basic machine learning tools in hyperbolic space including multinomial logistic regression, feed-forward and recurrent neural networks like gated recurrent units and LSTMs in order to embed sequential data and perform classification in hyperbolic space. They demonstrate empirical improvements on textual entailment and noisy-prefix recognition tasks using hyperbolic sentence embeddings. Cho et al. (2018) introduced a hyperbolic formulation for support vector machine classifiers and demonstrated performance improvements for multi-class prediction tasks on real-world complex networks as well as simulated datasets.

Zipf's law states that word-frequency distributions obey a power law, which defines a hierarchy based on semantic specificities. Concretely, semantically general words that occur in a wider range of contexts are closer to the root of the hierarchy while rarer words are further down in the hierarchy. In order to capture the latent hierarchy in the natural language, there has been several proposals for training word embeddings in hyperbolic space. Dhingra et al. (2018) trains word embeddings using the algorithm from Nickel & Kiela (2017). They show that resulting hyperbolic word embeddings perform better on inferring lexical entailment relation than Euclidean embeddings trained with skip-gram model which is a standard method for training word embeddings, initially proposed by Mikolov et al. (2013). Leimeister & Wilson (2018) formulated the skip-gram loss function in hyperboloid model of hyperbolic space and evaluated on the standard the intrinsic evaluation tasks for word embeddings such as similarity and analogy in hyperbolic space.

Finally, the popularity of hyperbolic embeddings has stimulated interest in descent methods suitable for hyperbolic space optimization. In addition to tools like Bonnabel (2013) and Zhang et al. (2016), Zhang & Sra (2016) offers convergence rate analysis for a variety of algorithms and settings for Hadamard manifolds. Enokida et al. (2018) proposes an explicit update rule along geodesics in a hyperbolic space with a theoretical guarantee on convergence, and Zhang & Sra (2018) introduces an accelerated Riemannian gradient methods.

Our work also touches on previous work on maximum distance scaling (MDS) and PCA-like algorithms in hyperbolic, spherical, and more general manifolds. MDS-like algorithms in hyperbolic space are developed for visualization in Walter (2004) and Lamping & Rao (1994). Embeddings into spherical or into hyperbolic space with a PCA-like loss function were developed in Wilson et al. (2014). General forms of PCA include Geodesic PCA (Huckemann et al., 2010) and principal geodesics analysis (PGA) (Fletcher et al., 2004). A very general study of PCA-like algorithms is found in Pennec (2018).

## C   MANIFOLD CONCEPTS AND PROOFS

Below, we include proofs of our results and further discuss manifold notions such as curvature.

## C.1 MEANS IN PRODUCT SPACES

We begin with Lemma 2, restated below for convenience.

**Lemma 2.** *Let $\mathcal{P}$ be a product of model spaces of total dimension $r$, $T = \{p_1, \ldots, p_n\}$ points in $\mathcal{P}$ and $w_1, \ldots, w_n$ weights satisfying $w_i \geq 0$ and $\sum_{i=1}^{n} w_i = 1$. Moreover, let the components of the points in $\mathcal{P}$, $p_{i|\mathbb{S}^j}$ restricted to each spherical component space $\mathbb{S}^j$ fall in one hemisphere of $\mathbb{S}^j$. Then, Riemannian gradient descent recovers the mean $\mu(T)$ within distance $\epsilon$ in time $O(nr \log \epsilon^{-1})$.*

*Proof.* Consider the squared distance $d^2(p, q)$ for $p, q \in M$ for a manifold $M$. Fix $q$. We denote the Hessian in $p$ by $H_{p,M}(q)$. Then, we have the following expressions for the Hessian of the squared distance of a sphere, derived in Pennec

$$H_{p,\mathbb{S}^r}(q) = 2uu^T + 2\frac{\theta \cos \theta}{\sin \theta}(\mathbb{I}_r - pp^T - uu^t),$$

where $\mathbb{I}_r$ is the identity matrix, $\theta = \mathrm{acos}(p \cdot q)$ is the distance $d_{\mathbb{S}^r}(p, q)$ and $u = (\mathbb{I}_r - pp^T)q/\sin \theta$. In Pennec, it is shown that the eigenvectors of $H_{p,\mathbb{S}^r}(q)$ are $0, 1$, and $\theta \cot(\theta)$; thus the Hessian is bounded and if $\theta \in [0, \pi/2]$, it is also positive definite (PD).

For hyperbolic space (under the hyperboloid model), the Hessian is

$$H_{p,\mathbb{H}^r}(q) = 2uu^T J + 2\theta \coth \theta (J + pp^T - uu^T)J.$$

Here $\theta = \mathrm{acosh}(-\langle p, q \rangle_*)$, $J$ is the matrix associated with the Minkowski inner product, i.e., $\langle p, q \rangle_* = p^T Jq$, and $u = \log_p(q)/\theta$. The $\log$ here refers to the logarithmic map. That is, if $q = \exp_p(v)$, then $v = \log_p(q)$. Moreover, exact expressions for the eigenvalues of $H_{p,\mathbb{H}^r}(q)$ in terms of $\theta$ imply that it is always bounded and PD.

The Hessian for Euclidean space is $H_{p,\mathbb{E}^r}(q) = 2\mathbb{I}_r$, which is also PD.

Now we can express the Hessian of the weighted mean. We write $H_{p,\mathcal{P}}$ for the Hessian of the weighted variance $\sum_{i=1}^{n} w_i d_{\mathcal{P}}^2(p, p_i)$ (recall that $\mu(p_1, \ldots, p_n) = \arg\min_p \sum_{i=1}^{n} w_i d_{\mathcal{P}}^2(p, p_i)$). We have, by the decomposability of the distance, that

$$\sum_{i=1}^{n} w_i d_{\mathcal{P}}^2(p, p_i) = \sum_{j=1}^{s} \sum_{i=1}^{n} w_i d_{s_j}^2(p, p_i) + \sum_{j=1}^{h} \sum_{i=1}^{n} w_i d_{h_j}^2(p, p_i) + \sum_{i=1}^{n} w_i d_E^2(p, p_i).$$

Taking the Hessian,

$$H_{p,\mathcal{P}} = \sum_{j=1}^{s} \sum_{i=1}^{n} w_i H_{p,\mathbb{S}^{s_j}}(p_i) + \sum_{j=1}^{h} \sum_{i=1}^{n} w_i H_{p,\mathbb{H}^{h_j}}(p_i) + 2n\mathbb{I}_e.$$

Now, by assumption, the spherical components for our points in each of the spheres, $p_{i|\mathbb{S}^j}$, fall within one hemisphere, and we may initialize our gradient descent (that is, our $p_0$) within this hemisphere. Then, the angle $\theta$ in each of the spherical distances is in $[0, \pi/2]$, so that the corresponding Hessians are PD.

Since each term in the sum is PD and the weights satisfy $w_i \geq 0$, with at least one positive weight, $H_{p,\mathcal{P}}$ is also PD. Moreover, these Hessians are bounded. Then we apply Theorem 4.2 (Chap. 7.4) in Udriste (1994)), which shows linear rate convergence, as desired. $\square$

## C.2 CURVATURE ESTIMATION

We discuss the notions of curvature relevant to our product manifold in more depth. We start with a high-level overview of various definitions of curvature. Afterwards, we introduce the formal definitions for curvature and apply them to the product construction.

**Definitions of Curvature**    There are multiple notions of curvature, with varying granularity. Some of these notions are suitable for working with manifolds abstractly (without reference to an ambient space, that is, intrinsic). Others, in particular older definitions pre-dating the development of

the formal mechanisms underpinning differential geometry, require the use of the ambient space. Gauss defined the first intrinsic notion of curvature, *Gaussian* curvature. It is the product of the *principal curvatures*, which can be thought of as the smallest and largest curvature in different directions.[4] Below we consider several such notions.

*Scalar* curvature is a single value associated with a point $p \in M$ and intuitively relates to the area of geodesic balls. Negative curvature means volumes grow faster than in Euclidean space, positive means volumes grow slower.

A more fine-grained notion of curvature is that of *sectional curvature*: it varies over all "sheets" passing through $p$. Note that curvature is inherently a notion of two-dimensional surfaces, and the sectional curvature fully captures the most general notion of curvature (the Riemannian curvature tensor). More formally, for every two dimensional subspace $U$ of the tangent space $T_pM$, the sectional curvature $K(U)$ is equal to the Gaussian curvature of the sheet $\mathrm{Exp}_p(U)$. Intuitively, it measures how far apart two geodesics emanating from $p$ diverge. In positively curved spaces like the sphere, they diverge more slowly than in flat Euclidean space.

The Ricci curvature of a tangent vector $v$ at $p$ is the average of the sectional curvature $K(U)$ over all planes $U$ containing $v$. Geometrically the Ricci curvature measures how much the volume of a small cone around direction $v$ compares to the corresponding Euclidean cone. Positive curvature implies smaller volumes, and negative implies larger. Note that this is natural from the way geodesics bend in various curvatures. The scalar curvature is in fact defined as an average over the Ricci curvature, giving the intuitive relation between scalar curvature and volume. It is thus also an average over the sectional curvature.

**Discrete Analogs of Curvature**    Discrete data such as graphs do not have manifold structure. The goal of curvature analogs such as $\xi$ is to provide a discrete analog of curvature which satisfies similar properties to curvature; we use this to facilitate choosing an appropriate Riemannian manifold to embed discrete data into. In this work, we focus on the sectional curvature, but discrete versions of other curvatures have been proposed such as the the Forman-Ricci (Weber et al., 2017) and Ollivier-Ricci (Ollivier, 2009) curvatures.

The input to the discrete curvature estimation from Section 3.2 is analogous to other discrete curvature analogs. For example, the Ricci curvature is defined for a point $p$ and a tangent vector $u$, and the coarse Ricci curvature is defined for a node $p$ and neighbor $x$ (Ollivier, 2011). Similarly, the sectional curvature is defined for a point and two tangent vectors, and $\xi$ is defined for a a node and two neighbors.

**Sectional Curvature in Product Spaces**    Now we are ready to tackle the question of curvature in our proposed product space. Let $M$ be our Riemannian manifold and $\mathcal{X}(M)$ be the set of vector fields on $M$. The curvature $R$ of $M$ assigns a function $R(X, Y) : \mathcal{X}(M) \to \mathcal{X}(M)$ to each pair of vector fields $(X, Y) \in \mathcal{X} \times \mathcal{X}$. For a vector field $Z$ in $\mathcal{X}(M)$, the function $R(X, Y)$ can be written

$$R(X, Y)Z = \nabla_Y \nabla_X Z - \nabla_X \nabla_Y Z + \nabla_{[X,Y]} Z.$$

Here $\nabla$ is the Riemannian connection for the manifold $M$, and $[X, Y]$ is the Lie bracket of the vector fields $X, Y$.

For convenience, we shall write the inner product $\langle R(X, Y)Z, T \rangle$ as $(X, Y, Z, T)$; this is the *Riemannian curvature tensor*. Then, the sectional curvature is defined as follows. Let us take $\mathcal{V}$ to be a two-dimensional subspace of $T_pM$ and $x, y \in \mathcal{V}$ be linearly independent (so that they span $\mathcal{V}$). Then, the sectional curvature at $p$ for subspace $\mathcal{V}$ is

$$K(x, y) := \frac{(x, y, x, y)}{\|x\|^2 \|y\|^2 - \langle x, y \rangle^2}. \tag{4}$$

The model spaces $\mathbb{S}, \mathbb{H}, \mathbb{E}$ are the spaces of constant curvature, where $K$ is constant for all points $p$ and 2-subspaces $\mathcal{V}$.

---

[4] The curvature of a curve can be found by considering the *osculating circles* which match it to second order.

For simplicity, suppose we are working with $M = M_1 \times M_2$; the approach extends easily for larger products. We write $x = (x_1, x_2)$ for $x \in T_p M$. Similarly, let $R_1, R_2$ be the curvatures and $K_1, K_2$ be the sectional curvatures of $M_1, M_2$ at $p$, respectively. Then the curvature tensor decomposes as

$$R(X, Y)Z = (R_1(X_1, Y_1)Z_1, R_2(X_2, Y_2)Z_2). \tag{5}$$

Our goal is to evaluate the sectional curvature $K((x_1, x_2), (y_1, y_2))$ for the product manifold $M$. We show the following, re-stated for convenience:

**Lemma 1.** *Let $M = M_1 \times M_2$ where $M_i$ has constant curvature $K_i$. For any $u, v \in T_p M$, if $K_1, K_2$ are both non-negative, the sectional curvature satisfies $K(u, v) \in [0, \max\{K_1, K_2\}]$. If $K_1, K_2$ are both non-positive, the sectional curvature satisfies $K(u, v) \in [\min\{K_1, K_2\}, 0]$. If $K_i < 0$ and $K_j > 0$ for $i \neq j$, then $K(u, v) \in [K_i, K_j]$.*

*Proof.* Let us start with the numerator of equation (4):

$$\begin{aligned}
(x, y, x, y) &= ((x_1, x_2), (y_1, y_2), (x_1, x_2), (y_1, y_2)) \\
&= \langle R((x_1, x_2), (y_1, y_2))((x_1, x_2)), (y_1, y_2) \rangle \\
&= \langle ((R_1(x_1, y_1)x_1), (R_2(x_2, y_2)x_2) \rangle (y_1, y_2) \rangle \\
&= \langle R_1(x_1, y_1)x_1, y_1 \rangle + \langle R_2(x_2, y_2)x_2, y_2 \rangle
\end{aligned}$$

Here, we used equation 5 in the third line.

Note that when $x_1, y_1$ are linearly independent, then $\langle R_1(x_1, y_1)x_1, y_1 \rangle = K_1(\|x_1\|^2 \|y_1\|^2 - \langle x_1, y_1 \rangle^2)$ by (4). Otherwise, this still holds since it is 0. So we can relate the above to $K_1, K_2$:

$$(x, y, x, y) = K_1(\|x_1\|^2 \|y_1\|^2 - \langle x_1, y_1 \rangle^2) + K_2(\|x_2\|^2 \|y_2\|^2 - \langle x_2, y_2 \rangle^2).$$

For convenience, we write $\alpha_i = \|x_i\|^2 \|y_i\|^2 - \langle x_i, y_i \rangle^2$ for $i = 1, 2$. Then the numerator is simply $K_1 \alpha_1 + K_2 \alpha_2$. Next, we consider the denominator of (equation 4):

$$\begin{aligned}
\|x\|^2 \|y\|^2 - \langle x, y \rangle^2 &= \|(x_1, x_2)\|^2 \|(y_1, y_2)\|^2 - \langle (x_1, x_2), (y_1, y_2) \rangle^2 \\
&= (\|x_1\|^2 + \|x_2\|^2)(\|y_1\|^2 + \|y_2\|^2) - (\langle x_1, y_1 \rangle + \langle x_2, y_2 \rangle) \\
&= \alpha_1 + \alpha_2 + \|x_1\|^2 \|y_2\|^2 + \|x_2\|^2 \|y_1\|^2 \\
&= \alpha_1 + \alpha_2 + \beta,
\end{aligned}$$

where we set $\beta = \|x_1\|^2 \|y_2\|^2 + \|x_2\|^2 \|y_1\|^2$. Thus, we have that

$$K((x_1, x_2), (y_1, y_2)) = \frac{\alpha_1 K_1}{\alpha_1 + \alpha_2 + \beta} + \frac{\alpha_2 K_2}{\alpha_1 + \alpha_2 + \beta}. \tag{6}$$

Now, note that $\beta > 0$, since we assumed that $x_1, y_1$ and $x_2, y_2$ are linearly independent. By Cauchy-Schwarz, $\alpha_i \geq 0$. Then, if $K_i \geq 0$, we have that $(\alpha_i K_i)/(\alpha_1 + \alpha_2 + \beta) \leq (\alpha_i K_i)/(\alpha_1 + \alpha_2)$, so that

$$0 \leq K((x_1, x_2), (y_1, y_2)) \leq \frac{\alpha_1}{\alpha_1 + \alpha_2} K_1 + \frac{\alpha_2}{\alpha_1 + \alpha_2} K_2. \tag{7}$$

Thus, we relate the product sectional curvature to a convex combination of the factor sectional curvatures $K_1, K_2$. We have for non-negative $K_1, K_2$ (e.g., Euclidean and spherical spaces) that $K((x_1, x_2), (y_1, y_2)) \in [0, \max\{K_1, K_2\}]$. A similar result holds for the non-positive (Euclidean and hyperbolic) case. The last case (one negative, one positive space) follows along the same lines. $\quad\square$

**Distribution of $K$**   The range of curvatures from Lemma 1 can be easily extended to a more refined distributional analysis. In particular, consider sampling any point $p$ and a random plane $\mathcal{V} \subseteq T_p M$. By homogeneity, we can equivalently fix $p$. The 2-subspaces of $T_p M \simeq \mathbb{R}^d$ forms the Grassmannian manifold $\mathbf{Gr}(2, T_p M)$. The uniform measure on this (i.e. invariant to multiplication by an orthogonal matrix) can be recovered from the Haar measure on the orthogonal group $\mathrm{O}(d)$, which

---

**Algorithm 2** Sectional curvature distribution

---

1: **Input: Dimensions** $d_1, d_2$
2: $a_1 \leftarrow \chi^2(d_1 - 1)$
3: $b_1 \leftarrow \chi^2(d_1 - 1)$
4: $t_1 \leftarrow \text{Beta}((d_1 - 1)/2, (d_1 - 1)/2)$
5: $c_1 \leftarrow a_1^{1/2} b_1^{1/2} (2t_1 - 1)$
6: $a_2 \leftarrow \chi^2(d_2 - 1)$
7: $b_2 \leftarrow \chi^2(d_2 - 1)$
8: $t_2 \leftarrow \text{Beta}((d_2 - 1)/2, (d_2 - 1)/2)$
9: $c_2 \leftarrow a_2^{1/2} b_2^{1/2} (2t_2 - 1)$
10: $\alpha_1 \leftarrow a_1 b_1 - c_1^2$
11: $\alpha_2 \leftarrow a_2 b_2 - c_2^2$
12: $\beta \leftarrow a_1 b_2 + a_2 b_1$
13: **return** $\frac{\alpha_1}{\alpha_1 + \alpha_2 + \beta} K_1 + \frac{\alpha_2}{\alpha_1 + \alpha_2 + \beta} K_2$

---

itself can be constructed by orthonormalizing independent random normal vectors. In particular, it suffices to consider $\mathcal{V}$ spanned by independent Gaussians $x, y \sim \mathcal{N}(0, I)$.

Furthermore, we do not actually need to sample $d$-dimensional vectors $x, y$ to compute the relevant curvature in equation 6. It suffices to sample the quantitities $\langle x_1, y_1 \rangle, \langle x_1, x_1 \rangle, \langle y_1, y_1 \rangle$ and $\langle x_2, y_2 \rangle, \langle x_2, x_2 \rangle, \langle y_2, y_2 \rangle$ directly. Note that $\alpha := \langle x_1, x_1 \rangle$ and $\beta := \langle y_1, y_1 \rangle$ are $\chi^2$-distributed, while $\langle x_1, y_1 \rangle \sim \sqrt{\alpha \beta} \gamma$, where $\gamma$ is distributed as the dot product of two uniformly random unit vectors. By rotational invariance, this is the same as the first coordinate of a random unit vector, which in turn is distributed as $X_1^2 / (X_1^2 + \cdots + X_d^2)$ for independent normal $X_i$, and therefore $(\gamma + 1)/2 \sim \text{Beta}((d - 1)/2, (d - 1)/2)$.

Thus a random $K(\mathcal{V})$ can be computed by sampling from well known distributions in constant time, via Algorithm 2.

Furthermore, without knowing $K_1, K_2$ a priori, an estimate for these curvatures can be found by matching the distribution of sectional curvature from Algorithm 2 to the empirical curvature computed from Algorithm 3. In particular, Algorithm 2 can be used to generate samples for $\frac{\alpha_1}{\alpha_1 + \alpha_2 + \beta}$ and $\frac{\alpha_2}{\alpha_1 + \alpha_2 + \beta}$. The overall moments are then simple functions of $K_1, K_2$, and the sample moments of the above quantities, so that $K_1, K_2$ can then be found by matching moments.

**Curvature Estimation**   We prove the facts we mentioned in the main body of the paper relating to the evaluation of $\xi$ over fundamental pieces of graphs: lines, cycles, and trees.

**Lemma 3.** *Suppose $a$ lies on the same geodesic line as $b, m, c$; in other words, WLOG $d_G(a, b) \leq d_G(a, c)$ and suppose $d_G(a, c) = d_G(a, b) + d_G(b, m) + d_G(m, c)$. Then $\xi(m; b, c; a) = 0$.*

**Lemma 4.** *Consider a cycle graph $C$ with nodes $b, m, c$ such that $(m, b)$ and $(m, c)$ are neighbors. Then for all $a \in C$, $\xi(m; b, c; a)$ is either $0$ or positive.*

*Proof.* Without loss of generality, let the cycle have an even number of vertices $n$. Let $k$ be the node diametrically opposite from $m$. Note that for any vertex $a \neq n$, $a, b, m, c$ lie on a geodesic line, and therefore $\xi(m; b, c; a) = 0$. On the other hand,

$$\xi(m; b, c; a) = \frac{1}{2 \cdot n/2} \left( \left( \frac{n}{2} \right)^2 + 1 - \frac{1}{2} \left( (n/2 - 1)^2 + (n/2 - 1)^2 \right) \right) = 1.$$

The case when $n$ is odd is similar, where we find that two nodes $a$ satisfy $\xi(m; b, c; a) = n/(n - 1)$ and the rest are $0$.

$\square$

**Lemma 5.** *Consider a tree graph $T$ with nodes $b, m, c$ such that $(m, b)$ and $(m, c)$ are neighbors. Then for all $a \in T$, $\xi(m; b, c; a)$ is either $0$ or negative.*

---

**Algorithm 3** Empirical estimation of sectional curvature distribution

---

1: **Input: Graph** $G = (V, E)$
2: $m \leftarrow \text{Uniform}(V)$
3: $b \leftarrow \text{Uniform}(\mathcal{N}(m))$ $\{\mathcal{N}(v) \text{ is the neighbor set of } v\}$
4: $c \leftarrow \text{Uniform}(\mathcal{N}(m))$
5: $a \leftarrow \text{Uniform}(V)$
6: $K \leftarrow \xi(m; b, c; a)$
7: **return** $K$

---

*Proof.* Due to the tree structure, $a$ is either geodesic with $b, m, c$, or $a$ is connected to $m$ with a path that does not pass through $b$ or $c$. In the former case, $\xi(m; b, c; a) = 0$. In the latter case,

$$\xi(m; b, c; a) = \frac{1}{2d} \left( d^2 + 1 - (d+1)^2 \right) = -1,$$

where $d = d_G(a, m)$. □

Given a graph, the distribution of $\xi(m; b, c)$ over random "planes" (i.e. pairs of neighbors $b, c$) is easily calculable. This yields a distribution that can then be averaged over $m$ to obtain an average sectional curvature distribution. To simplify this, we find the distribution via sampling (Algorithm 3) in the calculations for Table 3, before being fed into Algorithm 2 to estimate $K_i$.

As a corollary of Lemma 5, note that the $\xi$ becomes more negative for trees of higher degree, matching the intuition that higher degree trees are more appropriate for hyperbolic space embeddings (Sala et al., 2018).

We additionally note that a line of work has studied discrete analogs of curvature for regular objects such as triangular planar tessellations (including polyhedra) (Thurston, 1998). For example, these notions assign positive curvature to regular polyhedra (the tetrahedron, octahedron, icosahedron), zero curvature to the flat planar tessellation, and positive curvature to the order-7 triangular tiling of the hyperbolic plane. It is easily checked that Algorithm 3 assigns the right curvature sign to each of these objects.

### C.3 COMBINATORIAL CONSTRUCTIONS

The $\ell_1$ and $\min$-based distances are suitable for combinatorial constructions where we do not learn embeddings by optimizing a surrogate loss function, but rather by directly placing points in the product space, often via recursive procedures. Such constructions offer superior speed and other benefits. On the other hand, they are only available for certain classes of graphs. Additionally, since the constructions rely on the $\ell_1$ and $\min$ distances, they do not take advantage of the Riemannian structure, and thus do not have the same applicability downstream.

We often exploit the combinatorial construction for trees from Sala et al. (2018) as a building block; it offers worst-case distortion[5] $1 + \varepsilon$ when embedding a tree into $\mathbb{H}^r$ for all $r \geq 2$, where we can control $\varepsilon$.

**Hanging Tree Construction** Consider the class of graphs $G = (V, E)$ where $V = B \cup T_1 \cup T_2 \ldots \cup T_{|B|}$, so that $B$ is a base set of nodes and for each node $a \in B$, there is a tree $T_a$ connected to $a$ (the hanging trees). We show how to use $\mathcal{P}$ with the $\ell_1$ distance to reduce the cost of embedding $G$ to that of embedding the subgraph induced by $B$.

We embed $G$ into the product space $\mathcal{P} = \mathcal{P}' \times \mathbb{H}^r$ equipped with the $\ell_1$ distance. Here, $\mathcal{P}'$ is some product manifold. We do the embedding in two steps:

---

[5]The worst-case distortion $D_{\text{wc}}$ is a commonly-considered variant of distortion

$$D_{\text{wc}}(f) = \frac{\max_{u,v \in U : u \neq v} d_V(f(u), f(v))/d_U(u, v)}{\min_{u,v \in U : u \neq v} d_V(f(u), f(v))/d_U(u, v)}.$$

The worst-case distortion is the ratio of the worst expansion and the worst contraction of distances; note that it is scale-invariant. Here, the best worst-case distortion is $D_{\text{wc}}(f) = 1$.

1. Embed the subgraph induced by $B$ into $\mathcal{P}'$ by any method; let the resulting worst-case distortion be $1 + \delta$. Embed every node in $T_i$ into the embedded image of node $i$,

2. Form the tree $T$ by connecting each of the $T_1, \ldots, T_{|B|}$ to a single node (equivalent to crushing all the nodes in $B$ into a single node), and embed $T$ into $\mathbb{H}^r$ by using the combinatorial construction. Additionally, all of the nodes in $B$ are embedded into the image of the single central node in $\mathbb{H}^r$.

We can check the distortion. For nodes $x_a, y_b$ in subtrees hanging off $a, b \in B$, the distance is $d_G(x_a, y_b) = d_T(x_a, y_b) + d_B(x, y)$. Since the distortion for the two embeddings are given by $1 + \delta$ and $1 + \varepsilon$, it is easy to check that the overall distortion is at most $\max\{1 + \delta, 1 + \varepsilon\}$.

As a concrete example, consider the ring of trees in Figure 1. Then, $B = C_r$, the cycle on $r$ nodes. In this case, we can embed $B$ into $\mathcal{P}' = \mathbb{S}^1$. Let the nodes of $B$ be indexed $a_1, \ldots, a_r$. We embed $a_i$ into $A_i = (\cos(\frac{2\pi i}{d}), \sin(\frac{2\pi i}{d}))$. Then, for $i < j$,

$$
\begin{aligned}
d_S(A_i, A_j) &= \text{acos}(A_i \cdot A_j) \\
&= \text{acos}\left(\cos\left(\frac{2\pi i}{d}\right)\cos\left(\frac{2\pi j}{d}\right) + \sin\left(\frac{2\pi i}{d}\right)\sin\left(\frac{2\pi j}{d}\right)\right) \\
&= \text{acos}\left(\cos\left(\frac{2\pi(i-j)}{d}\right)\right) \\
&= \frac{2\pi(|j-i|)}{d}.
\end{aligned}
$$

Thus indeed, the embedding has worst-case distortion 1. Thus, the overall distortion for the ring of trees is $1 + \varepsilon$. Since we control $\varepsilon$, we can achieve arbitrarily good distortion for the ring of trees. The complexity of this algorithm is linear in the number of nodes, since embedding the trees and ring is linear time.

**General Graph Construction**  Now we use the min distance space to construct an embedding of *any* graph $G$ on $r$ nodes with arbitrarily low distortion via the space $\mathcal{P} = \mathbb{H}^2 \times \mathbb{H}^2 \times \ldots \times \mathbb{H}^2$ with $r - 1$ copies. As we shall see, this construction is ideal (arbitrarily low distortion, any graph) other than requiring $O(r)$ spaces.

Let the nodes of $G$ be $V = \{a_1, \ldots, a_r\}$. Now, for each $a_i$, $1 \le i \le r - 1$, form the minimum distance tree $T_i$ rooted at $a_i$. Then, embed $T_i$ into the $i$th copy of $\mathbb{H}^2$ via the combinatorial construction. Then, for any nodes $a_i, a_j \in V$, the distance $d_G(a_i, a_j)$ is attained by $d_{T_i}(a_i, a_j)$, or $d_{T_j}(a_j, a_i)$ in the case $i = r$. Since at least one of $T_i$ or $T_j$, say $T_i$, is embedded in $\mathbb{H}^2$ with distortion $1 + \varepsilon$, if we make $\varepsilon$ small enough, the smallest distance among the embedded copies is indeed that for $T_i$, so our overall distortion is still $1 + \varepsilon$.

## D  VISUALIZATIONS AND INTERPRETABILITY

The combinatorial construction using the $\ell_1$ distance (Section C.3) can embed the hanging tree graph arbitrarily well, unlike any single type of space. Unlike a single space, this also lends more interpretability to the embedding, as each component displays different qualitative aspects of the underlying graph structure. Figure 4 shows that this phenomenon does in fact happen empirically, even using the optimization approach over the $\ell_2$ (Riemannian) instead of $\ell_1$ distance.

## E  EXPERIMENTAL DETAILS

We provide some additional details for our experimental setups.

**Graph Reconstruction**  The optimization framework was implemented in PyTorch. The loss function (2) was optimized with SGD using minibatches of 65536 edges for the real-world datasets, and ran for 2000 epochs. For the Cities graph, the learning rate was chosen among

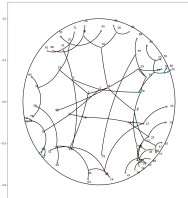 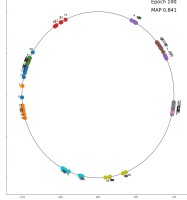 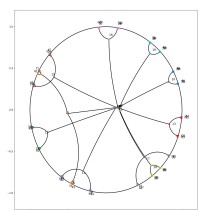 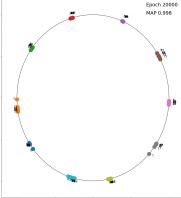

Figure 4: Ring of trees graph embedding into $(\mathbb{H}^2)^1 \times (\mathbb{S}^1)^1$; left: early epoch, right: completion. Only accessing graph distances, the optimization separates the different intrinsic structures of the underlying graph—the cycle and the trees—into interpretable components. Compare to Figure 2.

$\{0.001, 0.003, 0.01\}$. For the rest of the datasets, the learning rate was chosen from a grid search among $\{10, 30, 100, 300, 1000\}$ for each method.[6]

Each point in the embedding is initialized randomly according to a uniform or Normal distribution in each coordinate with standard deviation $10^{-3}$. (In the hyperboloid and spherical models, all but the first coordinate is chosen randomly, and the first coordinate is a function of the rest.)

Table 2 uses only Algorithm 1, and initializes the curvatures to $-1$ for hyperbolic components and $1$ for spherical components. These curvatures are learned using the method described in Section 3.1, and the "Best model" row reports the final curvatures of the best signature.

**Word Embeddings** Following LW, the input corpus is a 2013 dump of Wikipedia that has been preprocessed by lower casing and removing punctuation, and filtered to remove articles few page views. All other hyperparameters are chosen exactly as in as LW, including their numbers for Euclidean embeddings from *fastText*. The datasets used for similarity (WS-353, Simlex-999, MEN) and analogy (Google) are also identical to the previous setup.

---

[6]Note that the high LR stems from the particular choice of normalization for (2) in our implementation.

