# OpenReview forum: "Learning Mixed-Curvature Representations in Product Spaces"
_ICLR.cc/2019/Conference_

### Official Review · AnonReviewer2 · 2018-10-23
**Interesting ideas to explore towards understanding the geometry of data sets**

**Rating:** 7
**Confidence:** 3

**Review:**

The paper proposes a dimensionality reduction method that embeds data into a product manifold of spherical, Euclidean, and hyperbolic manifolds. The proposed algorithm is based on matching the geodesic distances on the product manifold to graph distances. I find the proposed method quite interesting and think that it might be promising in data analysis problems. Here are a few issues that would be good to clarify:

- Could you please formally define K in page 3?

- I find the estimation of the signature very interesting. However, I am confused about how the curvature calculation process is (or can be) integrated into the embedding method proposed in Algorithm 1. How exactly does the sectional curvature estimation find use in the current results? Is the “Best model” reported in Table 2 determined via the sectional curvature estimation method? If yes, it would be good to see also the Davg and mAP figures of the best model in Table 2 for comparison.

- I think it would also be good to compare the results in Table 2 to some standard dimensionality reduction algorithms like ISOMAP, for instance in terms of Davg. Does the proposed approach bring advantage over such algorithms that try to match the distances in the learnt domain with the geodesic distances in the original graph?

- As a general comment, my feeling about this paper is that the link between the different contributions does not stand out so clearly. In particular, how are the embedding algorithm in Section 3.1, the signature estimation algorithm in Section 3.2, and the Karcher mean discussed in Section 3.3 related? Can all these ideas find use in an overall representation learning framework?

- In the experimental results in page 7, it is argued that the product space does not perform worse than the optimal single constant curvature spaces. The figures in the experimental results seem to support this. However, I am wondering whether the complexity of learning the product space should also play a role in deciding in what kind of space the data should be embedded in. In particular, in a setting with limited availability of data samples, I guess the sample error might get too high if one tries to learn a very high dimensional product space.


Typos:

Page 3: Note the “analogy” to Euclidean products
Page 7 and Table 1: I guess “ring of cycles” should have been “ring of trees” instead
Page 13: Ganea et al formulates “basic basic” machine learning tools …

---

> ### Author Response · Authors · 2018-11-15
> **Thank you for your feedback**
>
> We appreciate the reviewer’s thoughtful feedback on our work.
>
> - On the definition of K
>
> K is a constant that parametrizes the curvature of the model spaces (hyperbolic, Euclidean, and spherical); for any constant K, there is a corresponding space with curvature K. In our notation, \mathbb{E}^d has curvature 0, \mathbb{S}^d_K has curvature K, and \mathbb{H}^d_K has curvature -K.
>
>
> - On the use of the signature estimation
>
> Table 2 does not use Algorithms 2 and 3, instead using Algorithm 1 with a variety of signatures to show the interaction between signature and dataset. For every experiment, the curvatures are initialized to -1,0, or 1 for H, E, and S components resp., and learned using the method described in Section 3.1; this is what is reported in the Best model. These details have been clarified in Appendix D.
>
> As the reviewer has correctly observed, Algorithm 1 can be initialized with the estimated signature from Algorithms 2 and 3, which saves on hyperparameter searching and computation time. Table 3 shows that this method would indeed choose the best signature among the two-component options.
>
>
> - On comparison vs ISOMAP
>
> We thank the reviewer for pointing out ISOMAP, a non-linear dimensionality reduction algorithm. We ran an experiment to compare against our proposed techniques. We first embedded the graphs from 4.1 into a higher (100) dimensional Euclidean space, than used ISOMAP to reduce the dimension to 10 in order to compare the average distortion against the product manifolds from Section 4.1. We saw a d_avg for PhD's/Facebook/Power Graph/Cities 0.4085 / 2.2295 / 0.4863 / 0.3711. We hypothesize that while ISOMAP can be good for dimensionality reduction for an already-good Euclidean embedding (with many dimensions), it does not perform as well as our technique for situations when the higher-dimensional Euclidean embeddings themselves have non-zero distortion---nor can it capture the mixed-curvature manifolds our approach offers.
>
>
> - On the link between the different contributions
>
> The operational flow is the following. We start with the data to be embedded. We
>
> (i) seek an appropriate space to embed it in (in order to get a high-quality representation). To find what this embedding space should be, we estimate the signature (Section 3.2). More concretely, we use Algorithm 3 to estimate the distribution of discrete curvature of the data and Algorithm 2 to find a matching product manifold. This yields the "signature", i.e., a the number of factors and each factor's type and dimension for our product manifold.
>
> We have now selected an embedding space, and we
>
> (ii) perform the embedding. This is done via Algorithm 1(RSGD) in Section 3.1.
>
> Now we have an embedding. There are many further tasks to be done with these representations. Perhaps the most fundamental is to take the mean of the representations for a subset of the data. Since our embeddings are into a product manifold, this requires a slightly more sophisticated approach; we
>
> (iii) compute this mean via the Karcher mean detailed in Section 3.3.
>
>
> - On the complexity of learning the product space and the limited data sample regime
>
> This is an excellent point. We point out that (1) Optimization in the sphere and hyperboloid has the same complexity up to a constant as in Euclidean space, so that the complexity of our product manifold proposal is roughly the same as using SGD to produce typical embeddings, as we simply use R-SGD on the factor spaces. (2) The heuristic for choosing a space is very cheap (i.e., Algorithms 2 and 3) compared to the main embedding procedure, and is better suited for simple products anyways, avoiding the sample complexity issue of a large search space. Indeed, we do not seek to embed into higher dimensional spaces: our approach shows good results with few dimensions in a product space.

---

### Official Review · AnonReviewer3 · 2018-11-01
**The problem studied in the paper is interesting. However, there are various mathematical and theoretical problems with the paper, some of which are mentioned below. In addition, the claims and novelty of the paper fall short in the provided methods and results.**

**Rating:** 7
**Confidence:** 5

**Review:**


Page 2: What are p_i, i=1,2,...,n, their set T and \mathcal{P}?

What is | | used to compute distortion between a and b?

Please fix the definition of the Riemannian manifold, such that M is not just any manifold, but should be a smooth manifold or a particular differentiable manifold. Please update your definition more precisely, by checking page 328 in J.M. Lee, Introduction to Smooth Manifolds, 2012, or Page 38 in do Cormo, Riemannian Geometry, 1992.

Please define \mathcal{P} in equation (1).

Define K used in the definition of the hyperboloid more precisely.

Please provide proofs of these statements for product of manifolds with nonnegative and nonpositive curvatures: “In particular, the squared distance in the product decomposes via (1). In other words, dP is simply the l2 norm of the component distances dMi.”

Please explain what you mean by “without the need for optimization” in “These distances provide simple and interpretable embedding spaces using P, enabling us to introduce combinatorial constructions that allow for embeddings without the need for optimization.” In addition, how can you compute geodesic etc. if you use l1 distance for the embedded space?

By equation (2), the paper focuses on embedding graphs, which is indeed the main goal of the paper. Therefore, first, the novelty and claims of the paper should be revised for graph embedding. Second, three particular spaces are considered in this work, which are the sphere, hyperbolic manifold, and Euclidean space. Therefore, you cannot simply state your novelty for a general class of product spaces. Thus, the title, novelty, claims and other parts of the paper should be revised and updated according to the particular input and output spaces of embeddings considered in the paper.

Please explain how you compute the metric tensor  g_P and apply the Riemannian correction (multiply by the inverse of the metric tensor g_P) to determine the Riemannian gradient in the Algorithm 1, more precisely.

Step (9) of the Algorithm 1 is either wrong, or you compute v_i without projecting the Riemannian gradient. Please check your theoretical/experimental results and code according to this step.

What is h_i used in the Algorithm 1? Can we suppose that it is the ith component of h?

In step (6) and step (8), do you project individual components of the Riemannian gradient to the product manifold? Since their dimensions are different, how do you perform these projections, since definitions of the projections given on Page 5 cannot be applied? Please check your theoretical/experimental results and code accordingly.

Please define exp_{x^(t)_i}(vi) and Exp(U) more precisely. I suppose that they denote exponential maps.

How do you initialize x^(0) randomly?

The notation is pretty confusing and ambiguous. First, does x belong to an embedded Riemannian manifold P or a point on the graph, which will be embedded? According to equation (2), they are on the graph and they will be embedded. According to Algorithm 1, x^0 belongs to P, which is a Riemannian manifold as defined before. So, if x^(0) belongs to P, then L is already defined from P to R (in input of the Algorithm 1). Thereby, gradient \nabla L(x) is already a Riemannian gradient, not the Euclidean gradient, while you claim that \nabla L(x) is the Euclidean gradient in the text.

Overall, Algorithm 1 just performs a projection of Riemannian or Euclidean gradient  \nabla L(x) onto a point v_i for each ith individual manifold. Then, each v_i is projected back to a point on an individual component of the product manifold by an exponential map.

What do you mean by “sectional curvature, which is a function of a point p and two directions x; y from p”? Are x and y not points on a manifold?

You define \xi_G(m;b,c) for curvature estimation for a graph G. However, the goal was to map G to a Riemannian manifold. Then, do you also consider that G is itself a Riemannian manifold, or a submanifold?

What is P in the statement “the components of
the points in P” in Lemma 2?

What is \epsilon in Lemma 2?

How do you optimize positive w_i, i=1,2,...,n?

What is the “gradient descent” refered to in Lemma 2?

Please provide computational complexity and running time of the methods.

Please define \mathbb{I}_r.

At the third line of the first equation of the proof of Lemma 1, there is no x_2. Is this equation correct?

If at least of two of x1, y1, x2 and y2 are linearly dependents, then how does the result of Lemma 1 change?

Statements and results given in Lemma 1 are confusing. According to the result, e.g. for K=1, curvature of product manifold of sphere S and Euclidean space E is 1, and that of E and hyperbolic  H is 0. Then, could you please explain this result for the product of S, E and H, that is, explain the statement “The last case (one negative, one positive space) follows along the same lines.”? If the curvature of the product manifold is non-negative, then does it mean that the curvature of H is ignored in the computations?

What is \gamma more precisely? Is it a distribution or density function? If it is, then what does (\gamma+1)/2 denote?

The statements related to use of Algorithm 1 and SGD to optimize equation (2) are confusing. Please explain how you employed them together in detail.

Could you please clarify estimation of K_1 and K_2, if they are unknown. More precisely, the following statements are not clear;

- “Furthermore, without knowing K1, K2 a priori, an estimate for these curvatures can be found by matching the distribution of sectional curvature from Algorithm 2 to the empirical curvature computed from Algorithm 3. In particular, Algorithm 2 can be used to generate distributions, and K1, K2 can then be found by matching moments.” Please explain how in more detail? What is matching moments?

- “we find the distribution via sampling (Algorithm 3) in the calculations for Table 3, before being fed into Algorithm 2 to estimate Ki” How do you estimation K_1 and K_2 using Algorithm 3?

- Please define, “random (V)”, “random neighbor m” and “\delta_K/s” used in Algorithm 3 more precisely.

---

> ### Author Response · Authors · 2018-11-15
> **References**
>
> [B] Bauer et al. Modern Approaches to Discrete Curvature. Lecture Notes in Mathematics
> [Ficken] Ficken, “The Riemannian and Affine Differential Geometry of Product-Spaces”, Annals of Math., 1939.
> [LW] Leimeister and Wilson. Skip-gram word embeddings in hyperbolic space.
> [Levy] Levy, "Symmetric Tensors of The Second Order Whose Covariant Derivatives Vanish", Annals of Math., 1926.
> [NK1] Nickel and Kiela. Poincaré embeddings for learning hierarchical representations.
> [NK2] Nickel and Kiela. Learning continuous hierarchies in the Lorentz model of hyperbolic geometry.
> [Ollivier1] Ollivier. Ricci curvature of Markov chains on metric spaces.
> [Ollivier2] Ollivier. A visual introduction to Riemannian curvatures and some discrete generalizations.
> [SDGR] Sala, De Sa, Gu, Ré. Representation tradeoffs for hyperbolic embeddings.
> [Sarkar] Sarkar. Low distortion Delaunay embedding of trees in hyperbolic plane.
> [TS] Turaga and Srivastava, Riemannian Computing in Computer Vision, Springer 2016.
> [WSJ] Weber, Saucan, and Jost. Characterizing complex networks with Forman-Ricci curvature and associated geometric flows.
> [WL] Wilson and Leimeister. Gradient descent in hyperbolic space.
> [ZS] Zhang and Sra. First-order methods for geodesically convex optimization.

---

> ### Author Response · Authors · 2018-11-15
> **Line-by-line response (3)**
>
> - How do you optimize positive w_i, i=1,2,...,n?
>
> By convention, the weights w_i are constants independent of the optimization. For example, to compute the standard Euclidean mean one would take w_i = 1/n for all i.
>
>
> - What is the “gradient descent” refered to in Lemma 2?
>
> The usual Riemannian gradient descent, since it is a manifold.
>
>
> - Please provide computational complexity and running time of the methods.
>
> The complexity of the Karcher mean algorithm is O(nr log epsilon^(-1)), as described on Page 2, PP 3, line 4. The convergence rate of RSGD is standard [ZS]: it converges to a stationary point with rate O(c/t), where c is a constant and t is the number of iterations. Algorithms 2 and 3 find good estimates of the corresponding distributions in a small number (~10^4) of samples; each sample requires constant time for both algorithms.
>
>
> - Please define \mathbb{I}_r.
>
> This is standard notation for the r x r identity matrix, but we have explicitly defined it now.
>
>
> - At the third line of the first equation of the proof of Lemma 1, there is no x_2. Is this equation correct?
>
> The second R_1(x_1, y_1)x_1 should be R_2(x_2, y_2)x_2, which follows from directly applying equation (5) to the previous line.
>
>
> - If at least of two of x1, y1, x2 and y2 are linearly dependents, then how does the result of Lemma 1 change?
>
> The result does not change.
>
>
> - Statements and results given in Lemma 1 are confusing. According to the result, e.g. for K=1, curvature of product manifold of sphere S and Euclidean space E is 1, and that of E and hyperbolic  H is 0. Then, could you please explain this result for the product of S, E and H, that is, explain the statement “The last case (one negative, one positive space) follows along the same lines.”? If the curvature of the product manifold is non-negative, then does it mean that the curvature of H is ignored in the computations?
>
> In the case of a product of E and H, the sectional curvature ranges in [-1,0]. The line “and similarly for K_1, K_2 non-positive” implies that in the non-positive case we have K(u,v) \in [min(K_1, K_2), 0], since everything is negated.
>
>
> - What is \gamma more precisely? Is it a distribution or density function? If it is, then what does (\gamma+1)/2 denote?
>
> \gamma is a random variable which is distributed as the dot product of two uniformly random unit vectors, as defined on the bottom of page 16. Hence (\gamma+1)/2 is a well-defined random variable.
>
>
> - The statements related to use of Algorithm 1 and SGD to optimize equation (2) are confusing. Please explain how you employed them together in detail.
>
> Equation (2) is a loss function from \mathcal{P}^n to \mathbb{R} where the embeddings x_i are variables, and can thus be optimized using RSGD (Algorithm 1) on each point simultaneously. This is the same approach taken in previous works [NK1, SDGR, NK2] for the case of single space embeddings.
>
>
> - On estimation of K_1, K_2 and matching moments
>
> Algorithm 2 and 3 both produce distributions. Moment matching (or the method of moments) is a standard term referring to parameter estimation via equating the moments of distributions. More details have been added to the revised draft.
>
>
> - Please define, “random (V)”, “random neighbor m” and “\delta_K/s” used in Algorithm 3 more precisely.
>
> We have clarified that the random sampling is uniform. \delta_K refers to the delta function.

---

> ### Author Response · Authors · 2018-11-15
> **Line-by-line response (2)**
>
> - Step (9) of the Algorithm 1 is either wrong, or you compute v_i without projecting the Riemannian gradient. Please check your theoretical/experimental results and code according to this step.
>
> There is a typo; the RHS should have v_i instead of h_i.
>
>
> - What is h_i used in the Algorithm 1? Can we suppose that it is the ith component of h?
>
> h_i refers to the coordinates corresponding to the i-th component or factor.
>
>
> - In step (6) and step (8), do you project individual components of the Riemannian gradient to the product manifold? Since their dimensions are different, how do you perform these projections, since definitions of the projections given on Page 5 cannot be applied? Please check your theoretical/experimental results and code accordingly.
>
> Each projection is within its component; the text mentions each component is handled independently. A subscript i has been added to the RHS of steps (6),(8).
>
>
> - Please define exp_{x^(t)_i}(vi) and Exp(U) more precisely. I suppose that they denote exponential maps.
>
> Exp denotes the exponential map as defined in Section 2. The image Exp(U) refers to the standard notation f(S) := {f(s) : s \in S} where S is a set.
>
>
> - How do you initialize x^(0) randomly?
>
> The initialization scheme depends on the application. An example of a standard initialization selects each coordinate of x^(0) either uniform or Gaussian with std on the order of 1e-2 to 1e-3 [NK1, LW], which is what we also use in our empirical evaluation. We have clarified this in Appendix D.
>
> - The notation is pretty confusing and ambiguous. First, does x belong to an embedded Riemannian manifold P or a point on the graph, which will be embedded? According to equation (2), they are on the graph and they will be embedded. According to Algorithm 1, x^0 belongs to P, which is a Riemannian manifold as defined before. So, if x^(0) belongs to P, then L is already defined from P to R (in input of the Algorithm 1). Thereby, gradient \nabla L(x) is already a Riemannian gradient, not the Euclidean gradient, while you claim that \nabla L(x) is the Euclidean gradient in the text.
>
> x is the manifold point to be optimized. The notation \nabla L(x) is defined to be the Euclidean gradient at the bottom of page 4 of the initial submission. Note that this is the gradient of the embedding into ambient space; this is standard as in [NK2, WL].
>
>
> - Overall, Algorithm 1 just performs a projection of Riemannian or Euclidean gradient  \nabla L(x) onto a point v_i for each ith individual manifold. Then, each v_i is projected back to a point on an individual component of the product manifold by an exponential map.
>
> That is correct.
>
> - What do you mean by “sectional curvature, which is a function of a point p and two directions x; y from p”? Are x and y not points on a manifold?
>
> As mentioned earlier in the section, sectional curvature is a function of a point p and two directions (i.e. tangent vectors) u,v. However, tangent vectors can be identified with points on the manifold via geodesics (i.e. through Exp). The way our discrete curvature estimation is described in this section is analogous to other discrete curvature analogs [B]. For example, the Ricci curvature is defined for a point p and a tangent vector u, and the coarse Ricci curvature is defined for a node p and neighbor x [Ollivier2].
>
>
> - You define \xi_G(m;b,c) for curvature estimation for a graph G. However, the goal was to map G to a Riemannian manifold. Then, do you also consider that G is itself a Riemannian manifold, or a submanifold?
>
> G is a graph and does not have manifold structure. The goal of \xi is to provide a discrete analog of curvature which satisfies similar properties to curvature and facilitates choosing an appropriate Riemannian manifold to embed G into. There are other similar notions of discrete curvature on graphs, for example the Forman-Ricci [WSJ] and Ollivier-Ricci [Ollivier1] curvatures.
>
>
> - What is P in the statement “the components of the points in P” in Lemma 2?
>
> It is the product manifold. We have changed it to \mathcal{P}.
>
>
> - What is \epsilon in Lemma 2?
>
> \epsilon refers to a desired tolerance within which to compute the solution, in this case the mean. This is also explicitly mentioned in the last line of the second to last paragraph of Section 1. This is standard notation for gradient descent-based rates.

---

> ### Author Response · Authors · 2018-11-15
> **Line-by-line response (1)**
>
> - Page 2: What are p_i, i=1,2,...,n, their set T and \mathcal{P}?
>
> This refers to an arbitrary set T containing points p_1,...,p_n on a manifold P, for which we wish to define a mean.
>
>
> - What is | | used to compute distortion between a and b?
>
> Absolute value
>
>
> - Please fix the definition of the Riemannian manifold, such that M is not just any manifold, but should be a smooth manifold or a particular differentiable manifold. Please update your definition more precisely, by checking page 328 in J.M. Lee, Introduction to Smooth Manifolds, 2012, or Page 38 in do Cormo, Riemannian Geometry, 1992.
>
> Yes, it is a smooth manifold, as specified in the first line of the “Product Manifolds” paragraph.
>
>
> - Please define \mathcal{P} in equation (1).
>
> \mathcal{P} is a product manifold.
>
>
> - Define K used in the definition of the hyperboloid more precisely.
>
> K is an arbitrary constant that indexes the curvature. This is described in the first paragraph of section “Learning the curvature”.
>
>
> - Please provide proofs of these statements for product of manifolds with nonnegative and nonpositive curvatures: “In particular, the squared distance in the product decomposes via (1). In other words, dP is simply the l2 norm of the component distances dMi.”
>
> The given statement is a standard fact about products of Riemannian manifolds: some classical references are [Levy] and [Ficken], although the result is stated directly in, e.g, [TS, pg. 81, eq. (4.19)].  Here is a sketch of the proof: first, the Levi-Civita connection on the manifold decomposes along the product components [DoCarmo Ex. 6.1]. This implies that the acceleration is 0 iff it is 0 in each component; in other words, geodesics in the product manifold decompose into geodesics in each of the factors. The distance function’s decomposition follows from the additivity of the Riemannian metric, i.e. |\dot{\gamma}(t)| = \sqrt{\dot{\gamma_1}(t)^2 + \dot{\gamma_2}(t)^2}.
>
>
> - Please explain what you mean by “without the need for optimization” in… In addition, how can you compute geodesic etc. if you use l1 distance for the embedded space?
>
> * We are referring to embedding algorithms that do not require optimizing a loss function via, for example, gradient descent. This concept is detailed in Appendix C.3. For example, the second paragraph on page 19 shows how to embed a cycle by explicitly writing down the coordinates of the points, with no optimization. Similarly, for hyperbolic space, the combinatorial construction previously studied in [Sarkar, SDGR] embeds trees in hyperbolic space without optimization.
> * Additionally, it is explicitly mentioned in the first line of the corresponding paragraph that the alternative distances proposed are meant to “ignore the Riemannian structure”, because many common applications of embeddings such as link prediction do not actually require Riemannian manifold structure, or related notions such as geodesics. Conversely, the motivation for the application in Section 4.2 is to show a task where manifold structure and geodesics are actually required, where the (Riemannian) product is effective.
>
>
> - By equation (2), the paper focuses on embedding graphs, which is indeed the main goal of the paper. Therefore, first, the novelty and claims of the paper should be revised for graph embedding. Second, three particular spaces are considered in this work, which are the sphere, hyperbolic manifold, and Euclidean space. Therefore, you cannot simply state your novelty for a general class of product spaces. Thus, the title, novelty, claims and other parts of the paper should be revised and updated according to the particular input and output spaces of embeddings considered in the paper.
>
> Our embedding technique is not limited to graphs, and indeed we perform word embeddings into product manifolds as described in Section 4.2. Graphs, however, are used as a standard metric for non-Euclidean embeddings [NK1, SDGR, NK2], and so we evaluate our approach on a variety of graphs in Section 4.1. The language of graphs is also convenient for stating some of our results, but not necessary, as described in Footnote 1.
>
> The three particular spaces are the standard spaces of constant curvature, which has been considered in previous work. Our claimed novelty is in combining these using the Riemannian product construction to perform efficient embeddings into mixed-curvature spaces, as stated in the abstract (3rd sentence), introduction (3rd paragraph), and many other places throughout.
>
>
> - Please explain how you compute the metric tensor  g_P and apply the Riemannian correction (multiply by the inverse of the metric tensor g_P) to determine the Riemannian gradient in the Algorithm 1, more precisely.
>
> This is standard, as in [NK1,NK2, WL]. The only place it is necessary for us is for the hyperbolic components in Step (9).

---

> ### Author Response · Authors · 2018-11-15
> **Addressing your concerns**
>
> We welcome the reviewer's detailed questions and suggestions on the technical presentation of our paper, and we appreciate the opportunity to improve it. To the best of our understanding, many of the reviewer's questions are addressed in the submitted draft, or pertain to standard notation and arguments. Nevertheless, we respond to the reviewer’s comments in detail below, clarifying ideas or pointing out specific lines where questions are answered.
>
> We sincerely hope that our response clarifies any potential notational confusions, and we look forward to further engaging in a substantial discussion on the overall merits of our work.
>
> All pages and lines referenced refer to the original submission.

---

### Official Review · AnonReviewer1 · 2018-11-05
**Solid Paper**

**Rating:** 7
**Confidence:** 2

**Review:**

This paper proposes a new method to embed a graph onto a product of spherical/Euclidean/hyperbolic manifolds. The key is to use sectional curvature estimations to determine proper signature, i.e., all component manifolds, and then optimize over these manifolds. The results are validated on various synthetic and real graphs. The proposed idea is new, nontrivial, and is well supported by experimental evidence.

---

> ### Author Response · Authors · 2018-11-15
> **Thank you for your feedback**
>
> We appreciate the reviewer’s positive comments about our work.

---

### Meta-Review · Area_Chair1 · 2018-12-17
**Novel framework for learning non-euclidean embeddings**

**Confidence:** 5
**Recommendation:** Accept (Poster)

**Metareview:**

This paper proposes a novel framework for tractably learning non-eucliean embeddings that are product spaces formed by hyperbolic, spherical, and Euclidean components, providing a heterogenous mix of curvature properties.  On several datasets, these product space embeddings outperform single Euclidean or hyperbolic spaces. The reviewers unanimously recommend acceptance.